# A case study on topsoil removal and rewetting for paludiculture: effect on biogeochemistry and greenhouse gas emissions from *Typha latifolia*, *Typha angustifolia* and *Azolla filiculoides*

Merit van den Berg[1]*, Thomas Gremmen[2]*, Renske J.E. Vroom[3], Jacobus van Huissteden[1], Jim Boonman[1], Corine J.A. van Huissteden[1], Ype van der Velde[1], Alfons J.P. Smolders[2,3], Bas P. van de Riet[2]

*These authors contributed equally to this work.

[1]Department of Earth and Climate, Vrije Universiteit Amsterdam, Amsterdam, 1081 HV, Netherlands
[2]B-WARE Research Centre, Nijmegen, 6525 ED, Netherlands
[3]Department of Aquatic Ecology & Environmental Biology, Radboud University, Nijmegen, 6525 AJ, Netherlands

*Correspondence to*: Thomas Gremmen (t.gremmen@b-ware.eu)

**Abstract.** Rewetting drained peatlands for paludiculture purposes is a way to reduce peat oxidation (and thus $CO_2$ emissions) while at the same time it could generate an income for landowners, who need to convert their traditional farming into wetland farming. The side effect of rewetting drained peatlands is that it potentially induces high methane ($CH_4$) emission. Topsoil

removal could reduce this emission due to the removal of easily degradable carbon and nutrients. Another way to limit $CH_4$ emission is the choice in paludiculture species. In this study we conducted a field experiment in the coastal area of the Netherlands, in which a former non-intensively used drained peat grassland is rewetted to complete inundation (water table ~+18 cm) after a topsoil removal of ~20 cm. Two emergent macrophytes with a high potential of internal gas transport (*Typha latifolia* and *Typha angustifolia*), and a free floating macrophyte (*Azolla filiculoides*) were introduced and intensive

measurement campaigns were conducted to capture $CO_2$ and $CH_4$ fluxes, soil and surface water chemistry. Greenhouse gas fluxes were compared to a high-productive peat meadow as reference site.

Topsoil removal reduced the amount of phosphorus and iron in the soil to a large extent. The total amount of soil carbon per volume stayed more or less the same. The salinity of the soil was in general high defining the system as brackish. Despite the topsoil removal and salinity, we found very high $CH_4$ emission for *T. latifolia* (84.8 g $CH_4$ m$^{-2}$ yr$^{-1}$), compared to the much

lower emissions from *T. angustifolia* (36.9 g $CH_4$ m$^{-2}$ yr$^{-1}$) and *Azolla* (22.3 g $CH_4$ m$^{-2}$ yr$^{-1}$). The high emission can be partly explained by the large input of dissolved organic carbon into the system, but it could also be caused by plant stress factors, like salinity level and herbivory. For the total $CO_2$ flux (including C-export), the rewetting was effective, with a minor uptake of $CO_2$ for *Azolla* (-0.13 kg $CO_2$ m$^{-2}$ yr$^{-1}$) and a larger uptake for the *Typha* species (-1.14 and -1.26 kg $CO_2$ m$^{-2}$ yr$^{-1}$ for *T. angustifolia* and *T. latifolia*, respectively) compared to the emission of 2.06 kg $CO_2$ m$^{-2}$ yr$^{-1}$ for the reference site.

*T. angustifolia* and *Azolla*, followed by *T. latifolia* seem to have the highest potential in reducing greenhouse gas emissions after rewetting to flooded conditions (-1.4, 2.9 and 10.5 t $CO_2$-eq. ha$^{-1}$ y$^{r-1}$, respectively) compared to a reference drained peatlands (20.6 t $CO_2$-eq ha$^{-1}$ yr$^{-1}$). When considering the total greenhouse gas balance, other factors like biomass use, and storage of topsoil after removal should be considered. Especially the latter could cause substantial carbon losses if not kept in

anoxic conditions. When calculating the radiative forcing over time for the different paludicrops, which includes the GHG fluxes and the carbon release from the removed topsoil, *T. latifolia* will start to be beneficial to reduce global warming after 93 years, compared to the reference site. For both *Azolla* and *T. angustifolia* this will be after 43 years.

## 1 Introduction

With the increasing demand to reduce greenhouse gas (GHG) emissions to meet the climate goals, rewetting of drained peatlands has gained attention as a promising measure. Worldwide, drained peatlands are responsible for 2-5% of the total anthropogenic GHG emission, and reducing these emissions therefore have potentially a large contribution in mitigating climate change (Bonn et al., 2016; Leifeld and Menichetti, 2018; Humpenöder et al., 2020). The Netherlands has 260,000 ha of drained peat (6% of the total land area), mainly in use for agriculture. This area emits around 5.6 Mt $CO_2$-eq per year, which is about 3% of the total national emission (Arets et al., 2020). Besides the undesired effect of peat oxidation on the climate, it also leads to land subsidence of about 0.8 cm per year (Hoogland et al., 2012; Van den Born et al., 2016). For a country below sea level and an increasing sea level rise in prospect, this gives an extra incentive to reduce peat oxidation. By elevating the water table and thus rewetting the drained peat, anoxic conditions could be restored. Rewetting 60% of the drained organic soils would turn the global land system into a net C sink by 2100, as opposed to a net C source as projected (Humpenöder et al., 2020). However, rewetting and the return of anoxic conditions could lead to an increase in methane ($CH_4$) emissions and land becomes less suitable for conventional agriculture.

The increase in $CH_4$ emission after rewetting depends on the type of ecosystem and weather conditions (Abdalla et al., 2016; Hemes et al., 2018), but can be very high especially for rewetted grassland fens where availability of fresh organic matter is high (Hahn-Schöfl et al., 2011; Abdalla et al., 2016; Franz et al., 2016). The amount of $CH_4$ that is emitted also depends on the water table height. With complete inundation, no oxygen is available anymore resulting in potentially high $CH_4$ production and low $CH_4$ oxidation. However, with water tables below the surface, much of the produced $CH_4$ will be oxidized again resulting in low(er) emissions (Haldan et al., 2022). If soils are completely inundated, (nutrient-rich) topsoil can be removed prior rewetting to minimise high $CH_4$ emissions after peat rewetting (Harpenslager et al., 2015; Huth et al., 2020; Quadra et al., 2023).

$CH_4$ has a much stronger radiative forcing than $CO_2$, making the trade-off between $CO_2$ reduction and $CH_4$ emission complex. The short lifetime of $CH_4$ in the atmosphere (compared to $CO_2$), causes the effect on global warming to be time dependent. Most commonly, a global warming potential (GWP) of 27 on a timescale of 100 years is used to estimate climate impacts of $CH_4$ (IPCC, 2021). The use of this GWP as static number can be questioned if temporal forcing dynamics are considered (Günther et al., 2020). Despite the discussion on the effect of $CH_4$ on different time scales, keeping $CH_4$ emissions as low as possible always results in the lowest impact on the climate. Vegetation type plays a crucial role in the amount of $CH_4$ that is emitted due to the species-specific influence on substrate input, oxidizing of the rhizosphere and gas transport pathways (Hahn

et al., 2015; Abdalla et al., 2016; Vroom et al., 2022; Bastviken et al., 2023). Therefore, management of rewetted peatlands can be directed towards a vegetation type or composition that results in the lowest $CH_4$ emission.

After rewetting, agricultural land loses its carrying capacity and conventional crops and grasses are no longer suitable to grow. A transition to (semi) natural wetland would therefore be an option, but an alternative where biomass can still be commercially used and which generates a direct income for the landowner, is paludiculture – the cultivation of wetland plants on rewetted

peat. Ideally, paludiculture should result in restoration of peat accumulation (Wichtmann and Joosten, 2007). There are different potentially suitable plant species for paludiculture (for a list see Abel and Kallweit, 2022). Cattail (*Typha* spp.) is a favourable option due to the high biomass production (Haldan et al., 2022) and diverse potential use as building material (De Jong et al., 2021), fodder (Pijlman et al., 2019) and biogas (Martens et al., 2021). Additionally, *Typha* has a high nutrient extraction capacity which could be helpful to improve water quality (Vroom et al., 2018). *Typha* is a genus of perennial

emergent macrophytes, of which *Typha angustifolia* (narrowleaf cattail) and *Typha latifolia* (broadleaf cattail) are native to Europe and common in shallow freshwater habitats such as wetlands and drainage ditches (Clements, 2022; Murphy, 2022). Their high aerenchyma content (>50% of internal leaf volume) and pressurised gas transport (Pazourek, 1977; Sebacher et al., 1985) allow them to thrive in anoxic sediments, but can also lead to high $CH_4$ emissions from the sediment to the atmosphere (Sebacher et al., 1985). Generally, vegetation increases $CH_4$ emission (Kankaala et al., 2003; Hendriks et al., 2010; Zhang et

al., 2019; Bastviken et al., 2023; Bodmer et al., 2024) with the most important reason the input of carbon substrate for methanogens in the system, and plant mediated $CH_4$ transport. However, oxygen transport to the rootzone also increases $CH_4$ oxidation, which in some cases leads to lower $CH_4$ emission, as was found for some *Typha* lab/mesocosms studies (Van der Nat et al., 1998; Vroom et al., 2018; Bansal et al., 2020).

A much less discussed species in the context of paludiculture is water fern (*Azolla filiculoides*). Since its introduction from the

85 Americas to Western Europe in the late 19[th] century (Pieterse et al., 1977; Sheppard et al., 2006), *Azolla* is widespread in eutrophic shallow waters such as drainage ditches. *Azolla* has several traits which potentially make it an interesting crop for cultivation on rewetted agricultural lands. Because of its symbiosis with N-fixating cyanobacteria (Peters and Meeks, 1989) it has a very high potential clonal growth rate in phosphate-rich water (Wagner, 1997; Van Kempen, 2013; Li et al., 2018). Furthermore, the high protein and lipid content make it especially suitable for food and biofuel processing (Miranda et al.,

2016; Brouwer et al., 2019), or biofertilizer (Bocchi and Malgioglio, 2010). Dense floating mats of *Azolla* have shown to decrease light and $O_2$ concentrations in the underlying surface water (Pinero-Rodríguez et al., 2021) potentially resulting in increased phosphate mobilisation from the sediment to the overlying water (Boström et al., 1988).

This study was set-up to investigate the potential GHG emission reduction by three paludiculture species (from now on referred to as paludicrops): *T. latifolia, T. angustifolia,* and *A. filiculoides*, compared to high-productive drained peat grassland (from

95 now on referred to as reference). We aim to answer the following research questions:

1. Can $CO_2$ emission reduction compensate increased $CH_4$ emission after peatland rewetting and introduction of the three paludicrops?
   o Which of the three paludiculture species has the highest potential in reducing GHG emissions?

2.   What is the effect of topsoil removal and different paludicrop cultivation on soil and water nutrient concentrations?

In this study we looked at $CO_2$ and $CH_4$ dynamics in a field experiment on rewetted peat with the three different paludicrops. The experiment was conducted on a former drained and non-intensively managed fen grassland in West-Netherlands. At this site constructed wetland basins were created and the three plant species were introduced in 2018/2019 and GHG ($CO_2$ and $CH_4$) fluxes, soil and water chemistry were monitored in 2020. The total GHG budget was compared to the reference site at 4 km distance where $CO_2$ fluxes were measured in the same year and yearly $CH_4$ flux estimated based on data from the previous year (2019).

## 2 Materials and methods

### 2.1 Site description and experimental set-up

The experimental site was located on a former drained and non-intensively managed fen grassland in West-Netherlands (52°26'13"N, 4°43'50"E, Figure 1). The study started in 2018, when first ~20 cm of the topsoil was removed and used to construct embankments for the paludiculture basins. Soil properties and soil chemistry was measured before (2017) and after rewetting and topsoil removal (2018). Our experiment was conducted in four small basins (23 x 43 m). Within this paper we included intensive greenhouse gas measurements for one treatment basin (basin 2) (Figure 1C) that consisted of a water table around 18 cm above the soil surface level and no slurry or fertilizer application.

The basins were split into three compartments (~200 $m^2$ for *Azolla*, ~430 $m^2$ for each *Typha* species) by vertical wooden walls. Each wall had a water inlet so water could flow passively from the inlet ditch into all three compartments (Figure 2). The last compartment relative to the inlet ditch contained an overflow. Together with a fixed water level in the inflow ditch, this resulted in an average water level of 17.5 (2.5 S.D.) cm above the sediment surface. Per compartment a paludicrop was planted/introduced: *Azolla filiculoides*, *Typha angustifolia* and *Typha latifolia* (Figure 1). The *Typha* species were partly planted as seedlings in autumn 2018 and partly end March 2019. *A. filiculoides* (hereafter referred to as '*Azolla*') was introduced in March 2020, by placing 950 g $m^{-2}$ fresh weight in the water. *Azolla* covered the water surface at 90-100% from May to August, after which it declined due to an infestation of the water fern weevil *Stenopelmus rufinasus*. After infestation, *Lemna* spp. gradually took over, until no *Azolla* was left in December 2020 (see coverage of both species in Figure A1, Appendix B).

A wooden boardwalk on poles ran through the centres of each compartment to minimize disturbance during the measurements and sampling. A floating PVC frame of 3x3 m was placed to contain *Azolla* and minimize plant loss by wind. To reduce disturbance as much as possible during greenhouse gas flux measurements in the *Typha* plots, three wooden frames were installed below the water table in each *Typha* compartment (Figure 2).

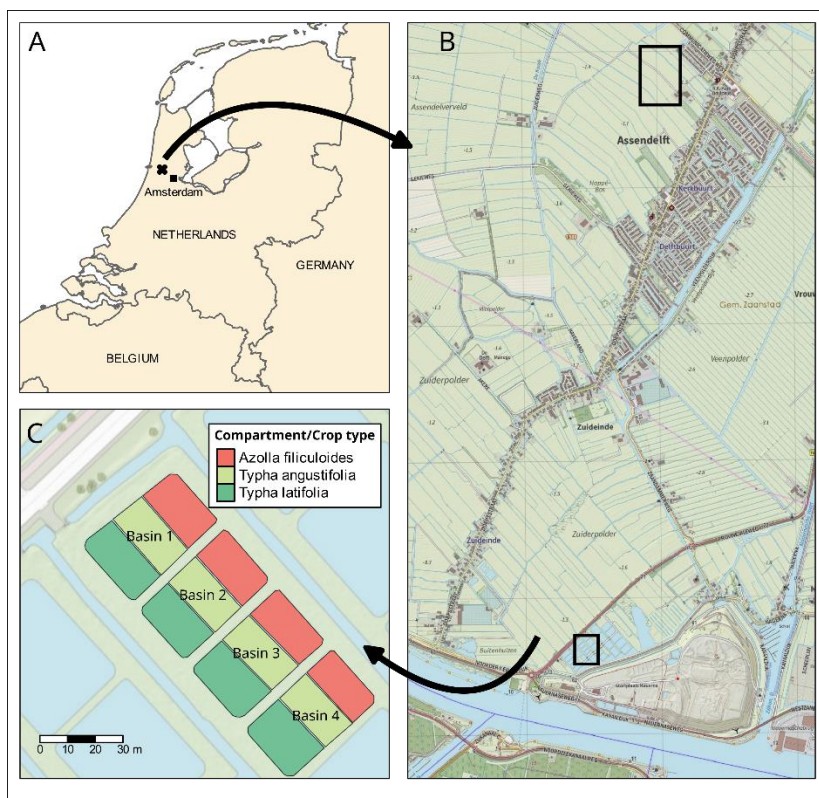

**Figure 1 A) Overview of the research area located in the Netherlands (source map: SPOTinfo), with in B) the paludiculture location in the small lower square and the reference drained fen grassland in the big upper square (source map: GADM). C) Measurements for this research were conducted in Basin 2. The other basins were used to test treatments that are not discussed in this paper.**

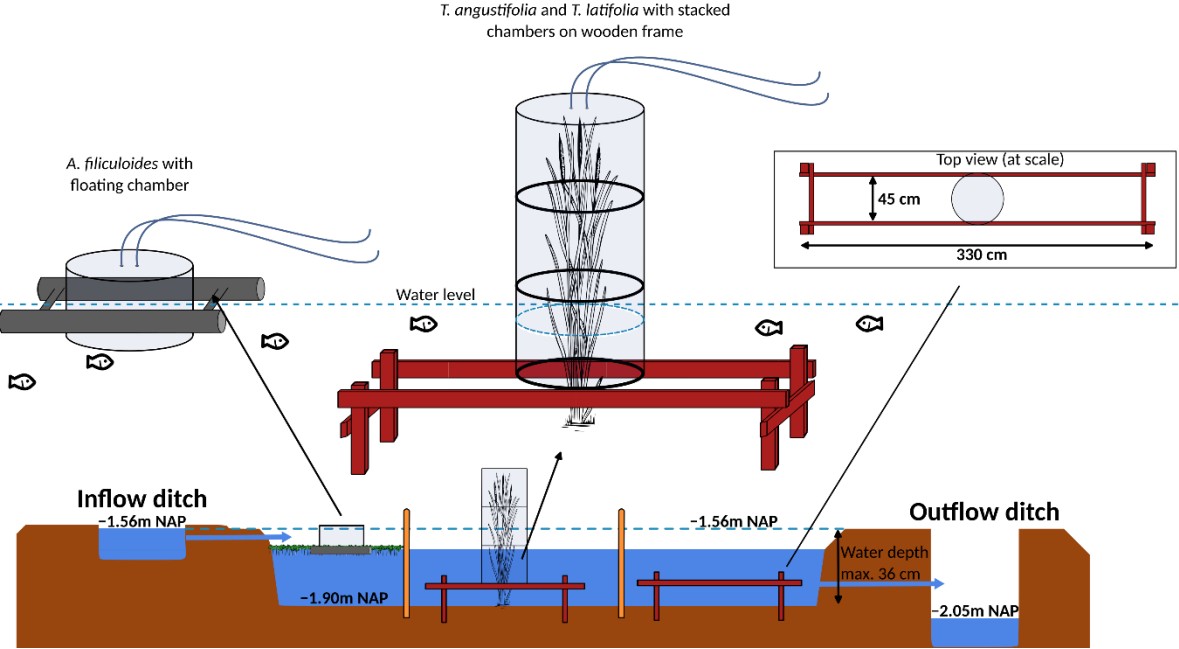

**Figure 2 Set-up greenhouse gas flux measurement in experimental basin. With on the left the overview for *Azolla* and in the middle and right for *Typha angustifolia* and *Typha latifolia*, respectively. Water tables are relative to Amsterdam Ordnance Datum (NAP).**

The reference site in Assendelft (52°28'31"N, 4°44'23"E) was a managed drained peatland used for dairy farming, where perennial ryegrass (*Lolium perenne*) is grazed and harvested during the growing season. Although the land management differs from the initial experimental site, they were similar in soil profile with a top layer (~30 cm) of clayey peat and several meters of peat below. In terms of water management both sites were also similar, with summer groundwater levels around 50 cm below surface. A large research plot of 24x10 m was fenced off, in which $CO_2$ flux measurements were done with automated

chambers and many environmental variables (like water table, soil- and air temperature, radiation) were monitored from April 2020 onwards. Chamber systems were relocated every two weeks at four different positions to minimize the effect of the chamber on temperature and vegetation growth. In 2020, every four weeks from 12 May to 29 October, grass was harvested from all chamber subplots and yield was determined at the latest chamber position and from a larger reference area next to the chambers (see Boonman et al., 2022). Fertilization was done with inorganic fertilizers (250 kg N, 108 kg $P_2O_5$ and 195 kg $K_2O$

$ha^{-1}$ $yr^{-1}$) to prevent carbon addition to the soil. Data from the reference site were gathered in the framework of a different project and more information about the site and measurements can be found in Boonman et al. (2022).

## 2.3 Flux measurements

In the experimental site, $CO_2$ and $CH_4$ fluxes were measured monthly from March-December 2020 with manual chambers, and five times (March, May, July, September/October) between the manual measurements with automated chambers aiming to capture diurnal patterns (results are described in Vroom et al., Under review).

For manual chamber measurements in both *Typha* species, transparent Perspex chambers (diameter 50 cm) were used that could be stacked to match the height of the plants (Figure 3). The chambers were equipped with a fan powered by a battery. The top part additionally contained a temperature logger and photosynthetically active radiation (PAR) logger (both HOBO onset, Onset Computer Corporation, Bourne, MA, USA). For *Azolla*, a floating transparent Perspex chamber was used (diameter 29 cm, height 26 cm) equipped with a HOBO temperature logger. PAR was measured with a handheld device outside the chamber (PAR Quantum sensor SKP 215, Skye Instruments, Llandrindod, Wales, UK). Chambers were connected in a closed loop with gastight tubing to either a LI-COR 7810 portable GHG analyser (LI-COR Inc, Lincoln, NE, USA) or a Los Gatos Ultra-Portable GHG analyser (ABB - Los Gatos Research, San Jose, CA, USA) that measured $CO_2$, $H_2O$ and $CH_4$ concentrations every second. Measurements were carried out during daytime and lasted three minutes each. Fluxes were alternated between light, darkened (chambers covered with opaque white plastic film) and shaded (chambers covered with plastic shading net, reducing PAR with ~42%) measurements, to cover the PAR range as much as possible. Per measurement campaign, three light, three darkened and three shaded measurements were done per species in three replicate locations (total n = 27). The increase in $CO_2$ and $CH_4$ concentrations in the chamber were visually checked for linearity in the field to ensure no ebullition occurred during the measurements. Measurements were redone if high concentration peaks, caused by ebullition, were detected. Fluxes were calculated by taking the linear fit of the concentration change in the first 1-3 minutes after closing the chamber.

Automated chambers consisted of four Perspex chambers with a diameter 35 cm and a height of 50 cm. The chambers can be extended up to 150 cm in height to match vegetation height (Figure 1) and were placed in one vegetation type at the time, measuring three days per vegetation type. These chambers were equipped with fans and DS18b50 temperature sensors. Furthermore, they had hinged lids controlled by a connected Raspberry Pi computer (Raspberry Pi Foundation, Cambridge, UK) and were connected with gastight tubing in three closed loops to a Los Gatos Microportable Greenhouse Gas Analyzer (ABB - Los Gatos Research, San Jose, CA, USA). The four chambers were measured in succession, by closing the lid of a respective chamber for 2.5 minutes, followed by one minute of flushing the chamber and gas analyser with atmospheric air, and then closing the lid of the next chamber, measuring, and flushing. This sequence continued for three days in each vegetation type, providing high resolution data including diurnal variation in emissions.

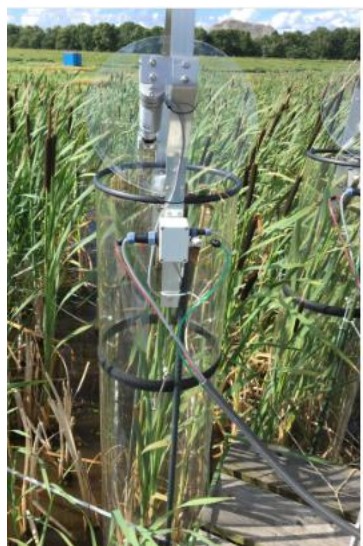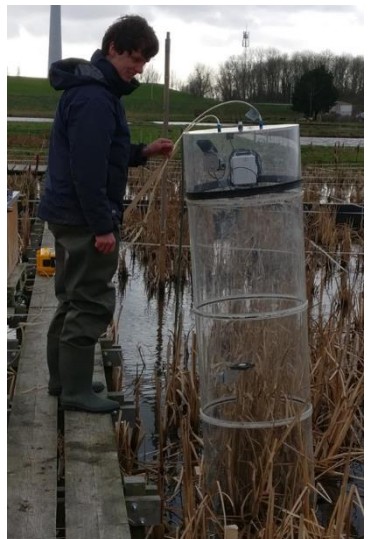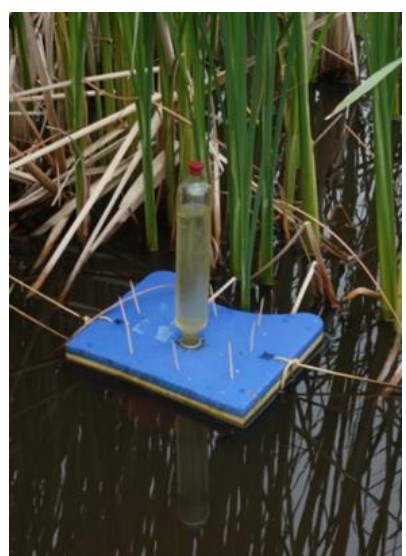

**Figure 3 Automated chamber (left), manual chamber (middle), and bubble trap (right), used for measuring fluxes.**

From March till December, ebullitive $CH_4$ fluxes were captured with bubble traps (see Aben et al., 2017) from the three

paludicrops. Three bubble traps were installed in each vegetation type (total n = 9). These traps consisted of a small floating foam raft with inserted a funnel (diameter 20 cm) on the bottom part connected to a glass tube above the raft (Figure 3). To prevent ducks from sitting on the raft, toothpicks were inserted in the foam. A butyl stopper was fitted at the top of each glass tube to enable gas extraction. Gas volume was determined every 1.5-3 weeks by removing the captured gas with a syringe. To determine $CH_4$ concentrations, gas was sampled once (April, May, November, December) or twice (June – October) a month.

$CH_4$ concentrations of 1 ml of diluted gas sample was measured with a Los Gatos Ultraportable GHG analyser using an open loop of gastight tubing.

CO_2 fluxes in the reference site (Assendelft) were measured with four automated chambers, connected in a closed loop to a LI-850 $CO_2$ gas analyzer (LI-COR, USA). The chambers had a height of 0.5 m and a diameter of 0.4 m. Every 15 minutes, each chamber measured for 3 minutes. More details about this chamber set-up and measurements can be found in Boonman et

al. (2022). $CH_4$ fluxes were not measured in 2020. However, in the year 2019 $CH_4$ fluxes were captured with the same method and frequency as described for the paludiculture plots. These data showed that $CH_4$ fluxes were close to zero on a yearly basis (-0,04 t $CO_2$-eq. ha$^{-1}$ yr$^{-1}$; Gremmen et al., 2022). Therefore, we assumed zero $CH_4$ emissions in 2020.

No flux measurements were done in ditches; therefore, the data only represents fluxes from the (rewetted) land area.

## 2.4 Partitioning and interpolation of fluxes

Measured net ecosystem exchange for $CO_2$ (NEE) from manual chambers and automated chambers averaged over 30 minutes were partitioned into gross primary production (GPP) and ecosystem respiration ($R_{eco}$) by subtracting the $R_{eco}$ based on dark measurements from the shaded/light measurements. With this the Lloyd-Taylor function (Lloyd & Taylor, 1994, eq. 1) for $R_{eco}$ and the light response curve for GPP could be fitted. The obtained parameters were used for interpolation of GPP and $R_{eco}$ between the measurement campaigns. The Lloyd-Taylor function was defined as:

$$R_{eco} = R_{ref} \times e^{E_0 \times \left( \frac{1}{T_{ref} - T_0} - \frac{1}{T - T_0} \right)}$$ (1)

where $R_{ref}$ = respiration at reference temperature ($T_{ref}$); $E_0$ = long term ecosystem sensitivity coefficient; $T_0$ = base temperature between 0 and T (227.13 K, Lloyd & Taylor, 1994); T = observed temperature (K); $T_{ref}$ = reference temperature (K).

$E_0$ was determined by fitting $E_0$ and $R_{ref}$ using the entire dataset, where $T_{ref}$ of 283.15 K was used, T was represented by average soil temperature at 5 cm depth, and the observed $R_{eco}$ were averaged dark or night-time $CO_2$ fluxes per measurement campaign/day. With the gained $E_0$, $R_{ref}$ per measurement campaign/day at a $T_{ref}$ of 283.15 K was determined by inverse the Lloyd-Taylor function with the average measured dark or night-time $CO_2$ flux as $R_{eco}$, and the average soil temperature at 5 cm depth as T. $R_{ref}$ was linearly interpolated between the measurement campaigns. Both $R_{ref}$ and $E_0$ were used to calculate $R_{eco}$ for every 30 minutes with measured soil temperature when data was absent.

Daytime fluxes were partitioned based on the standard procedure as used in e.g. Falge et al. (2001), Veenendaal et al. (2007) and Tiemeyer et al. (2016). GPP was gained from the NEE, by subtracting calculated $R_{eco}$: GPP = NEE – $R_{eco}$. The parameters $\alpha$ and $GPP_{max}$ of a hyperbolic light response equation based on the Michaelis–Menten kinetic (eq. 2), were fitted on the given GPP.

$$GPP = \frac{(\alpha \cdot PAR \cdot GPP_{max})}{(\alpha \cdot PAR + GPP_{max})}$$ (2)

where $\alpha$ is the initial slope of the light response curve; $GPP_{max}$ is the light-saturated photosynthetic rate and PAR is the measured photosynthetically active radiation. $GPP_{max}$ and $\alpha$ were linearly interpolated between the measurement campaigns and used to calculate GPP on 30-minute base when no data were present.

$CO_2$ partitioning and gapfilling (interpolation) was slightly different for the automated chambers on the reference location Assendelft, since the data density was much higher and $R_{eco}$ was determined from night-time data and calculated for daytime based on the temperature response of the Lloyd-Taylor relation. For more details see (Boonman et al., 2022). The only difference is the missing data from January-March 2020, since measurements started in April 2020. $R_{eco}$ for this period was estimated by fitting the Lloyd-Taylor function on the winter period January-March for the years 2021-2023 and using the

gained $R_{ref}$ and $E_0$ with the measured soil temperature for the period January-March 2020. To estimate GPP in 2020, we used the monthly average of the parameter α and the $GPP_{max}$ for January – March for 2021, since first harvest of 2020 and 2021 were similar, together with measured PAR in January-March 2020.

For $CH_4$, there is no standard interpolation procedure, and therefore to find to best relation between environmental variables and diffusive fluxes was part of this study. Many studies show that temperature is one of the best predictor variables for $CH_4$ fluxes (Kroon et al., 2010; Turetsky et al., 2014; Irvin et al., 2021). Other mentioned predictors are water table (which is not fluctuating in our study, and therefore not relevant) and vegetation. We choose to only use temperature to interpolate our data, and base that on the same principle as the interpolation of $R_{eco}$ with the Lloyd-Taylor function: we assume that influences of other factors than temperature, like vegetation, are captured in the mean measured $CH_4$ flux every 2-3 weeks and are linearly changing over time. We found that the best predictor variable for $CH_4$ fluxes was soil temperature for *Typha*, and water temperature for *Azolla,* with an exponential relation (see section 3.3 Methane fluxes). We used the mean measured $CH_4$ flux per campaign, calculated this back to a reference temperature of 10 °C with the gained temperature relation (Figure 7), linearly interpolated these reference emissions and calculated the actual emission with the measured temperature and the temperature relation from this reference emission.

## 2.5 Vegetation measurements

To estimate the biomass of *Typha* spp. per measurement campaign, average biomass per plant height was related to number of plants and plant height per campaign. For this, ten living shoots were harvested for each species outside of the measurement plots in September 2020. These shoots were dried at 70 °C for 72 hours, and dry weight per cm biomass was calculated. This was then multiplied by the number of living shoots and the average shoot height in each subplot to estimate the biomass at each measurement campaign (Figure A1). For the C-export term, the average number of dead stems was subtracted from the number of living stems from the measurement plots, and with the above-described relation used to determine the extra amount of biomass produced in 2020. This was called C-export since this could have been the potential net term of carbon loss by harvest.

For *Azolla*, biomass was not directly estimated, only the biomass cover per measurement campaign (Figure A1).

## 2.6 Sample collection

Soil samples of the experimental site before topsoil removal and rewetting were collected in March 2017. Samples were collected at five locations, divided over the area where the four experimental paludiculture basins were planned, at a depth of 0-10, 10-20 and 20-30 cm below surface level. After topsoil removal and rewetting, additional samples were collected in November 2018 in the four experimental basins. Here, five samples of the inundated topsoil (0-10 cm) were collected in each compartment of the four basins, after which the samples were pooled per compartment before analysis. All samples were stored in airtight plastic bags at 4 °C until further analysis.

Surface water and pore water samples were collected monthly directly after manual chamber measurements in the experimental basins and inflow- and outflow ditches. Surface water samples were taken by hand in the inlet water ditch and each compartment (n=1 per compartment/ditch). Pore water was collected anaerobically with a 60 ml syringe attached to a ceramic cup via gas-tight tubing, which was installed in the top 15 cm of the sediment in each respective compartment. Additional pore water samples for dissolved $CH_4$ and sulphide ($H_2S$) were collected by attaching a gas-tight pre-vacuumed 12 ml glass exetainer (containing 1 ml of 0.5M HCl to stop microbial activity; Labco, Lampeter, UK) via a hypodermic needle to the gas-tight tubing. The exetainers were stored upside down to minimise the risk of gas leakage. All samples were stored at 4 °C until further analysis.

## 2.7 Soil analysis

Two aluminium cups (40.5 ml) were filled with fresh soil and weighed before and after drying at 60 °C for >48 hrs to obtain the wet weight and bulk density, respectively. Thereafter, one cup with dried soil was incinerated (4 hrs at 550 °C) and weighed again to determine organic matter (as loss on ignition). Total phosphorus (P), iron (Fe) and sulphur (S) content were determined by digesting 200 mg of homogenised finely ground soil with 5 ml 65% $HNO_3$ and 2 ml $H_2O_2$ in a microwave (Ethos Easy, Milestone, Sorisole, Italy). Samples were then diluted to 100 ml with demineralised water and analysed using inductively coupled spectrometry axial plasma observation, seaspray nebulizer, 1300 W, 12 l min$^{-1}$ (ICP-OES ARCOS MV, Spectro Analytical Instruments, Kleve, Germany). Plant-available extractable inorganic nitrogen was determined by incubating 17.5 g of fresh soil with 50 ml of 0.2M NaCl for 2 hrs with 105 rpm and at room temperature. After determining the pH (PHC101 probe connected to HQ440d, Hach, Düsseldorf, Germany), the extract was collected using soil moisture samplers (Rhizon SMS, Eijkelkamp, Giesbeek, Netherlands), and analysed colourimetrically for nitrate ($NO_3^-$) and ammonium ($NH_4^+$) on a Seal auto-analyser III using hydrazine sulphate and salicylate reagent, respectively. Plant-available extractable phosphorus (Olsen-P) was determined by incubating 3 g of dried homogenised finely ground soil with 60 ml 0.5 M $NaHCO_3$ at pH 8.5 for 30 min with 105 rpm at room temperature. The pH of the medium was adjusted before incubation by adding NaOH when necessary. The extracted medium was diluted ten times with demineralised water and stored at 4°C until further analysis on ICP-OES as described above.

## 2.8 Surface water and pore water analysis

The pH was measured using a standard Ag/AgCl2 electrode connected to a Radiometer (Copenhagen, Type TIM840). The total inorganic carbon (TIC) concentration was determined by injecting a known amount of sample into an infra-red gas analyser (ABB Advance Optima IRGA), after which the concentrations of $CO_2$ and $HCO_3^-$ were calculated based on the pH equilibrium. Concentrations of nitrate ($NO_3^-$) and ammonium ($NH_4^+$) were determined colourimetrically on an auto-analyser as described above. Chloride ($Cl^-$) and phosphate ($PO_4^{3-}$) concentrations were determined colourimetrically on a Bran+Luebbe auto-analyser III system using respectively mercury(II)cyanide and ammonium molybdate/ascorbic acid as the reagent. Acidified samples (0.1 ml 65% $HNO_3$) were analysed for total-Fe, total-P and total-S on ICP-OES as described above. After

equilibrating to atmospheric pressure with $N_2$ gas, the concentrations of methane ($CH_4$) and sulphide ($S^{2-}$) were measured in the headspace of the exetainers by injecting into a 7890B gas chromatograph (Agilent Technologies, Santa Clara, USA) equipped with a Carbopack BHT100 glass column (2 m, ID 2 mm), flame ionization (FID) and flame photometric detector (FPD). Concentrations of dissolved organic carbon (DOC) were measured on a TOC-L CPH/CPN analyser (Shimadzu) after acidification with HCl to remove DIC.

## 2.9 Environmental variables

Surface water and groundwater levels were calculated based on hourly measurements of atmospheric pressure (Baro-Diver, Eijkelkamp, Giesbeek, Netherlands) and water pressure at a known depth (Cera-Diver, Eijkelkamp, Giesbeek, Netherlands). Air temperature was also measured hourly (Baro-Diver, Eijkelkamp, Giesbeek, Netherlands). Soil and water temperature was monitored with a 2-minute interval (HOBO S-TMB temperature probe connected to H21-USB station, Onset, Bourne, MA, USA). Water temperature was measured in the *T. angustifolia* compartment, and soil temperature was measured at 5 cm depth in each compartment. PAR was monitored with a 2 min interval at 3 m above water level using a HOBO S-LIA-M003 PAR sensor connected to a H21-USB station (Onset, Bourne, MA, USA).

## 3. Results

### 3.1 Environmental conditions

The study was conducted in the year 2020, which was a slightly warmer (+0.9 °C) year than the 10-year average 2001-2020 (Royal Netherlands Meteorological Institute KNMI). The yearly average precipitation was very similar to average (862 mm in 2020) but the summer period (Jun-Sep) was dryer (337 mm vs 474 mm).

The seasonal dynamics are clearly visible in all variables. The groundwater table of the reference site reaches a minimum of -86 cm in August and was on average -38 cm in 2020 (Figure 4). The water table in the paludiculture basin was kept more or less constant at +18 cm. Due to this water layer, the soil temperature fluctuations in the paludiculture basin were much more dampened than within the reference site (Figure 4).

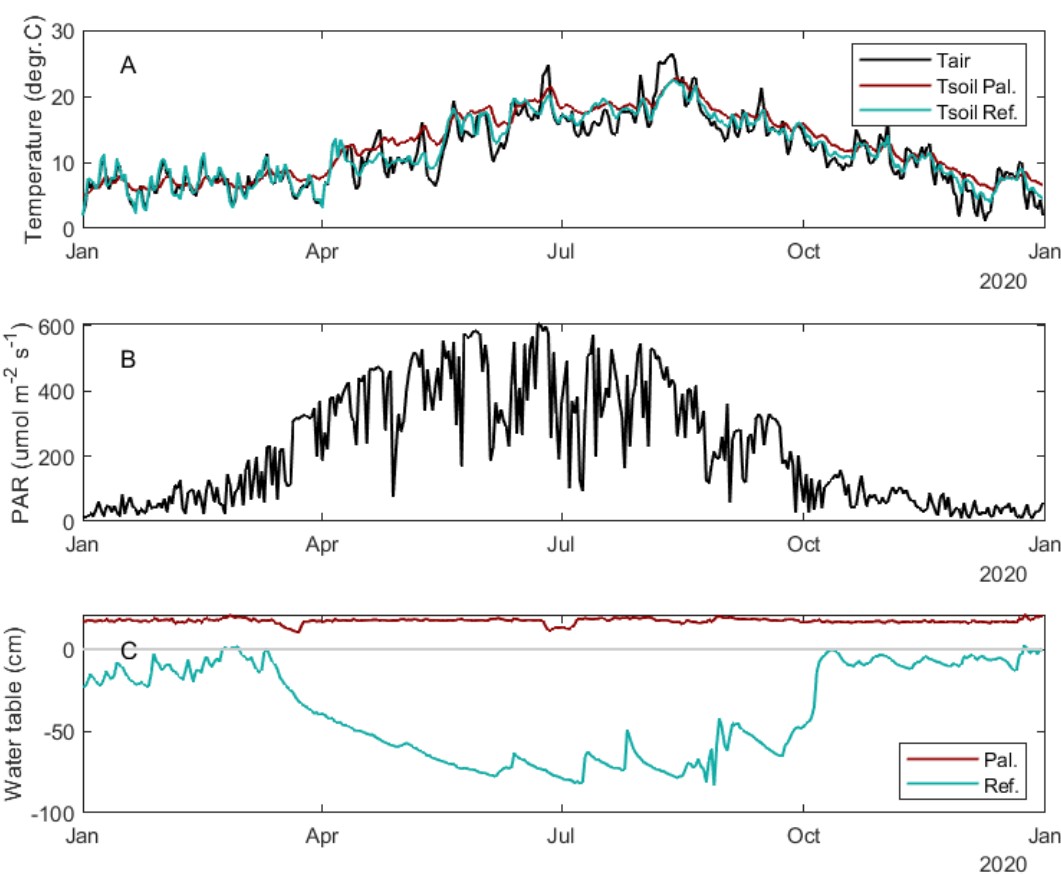

**Figure 4 (A) Air temperature (Tair) and site-specific soil temperature (Tsoil) for paludiculture site (Pal.) and reference site (Ref.), (B) photosynthetically active radiation (PAR), and (C) (ground)water table for the paludiculture site and reference site.**

## 3.2 Effect of rewetting and plant growth on surface and pore water chemistry

After topsoil removal, the total amount of organic matter remained the same in the upper soil layer, but bulk density in the top layer (0-10 cm) was reduced by about 50% (Table 1). Also, total phosphorus and total iron decreased quite drastically by about 65%, whereas total sulphur increased with 81%, which could lead to an increase in (toxic) free sulphide in the rootzone (Table 1). Assuming 58% of OM was carbon (C) (van Bemmelen factor), the removal of ~20 cm of the topsoil resulted in the displacement of approximately 15.8 kg C m$^{-2}$.

**Table 1** Soil properties of bulk density (BD), organic matter content (OM), total phosphorus (Total-P), plant-available phosphorus (Olsen-P), total iron (Total-Fe), and total sulphur (Total-S) before rewetting (RW) and topsoil removal in 2017 (n=5), and after rewetting and topsoil removal in 2018 (n=12). Because ~20 cm of topsoil is removed, the depth 0-10 cm after RW correspond to the soil layer 20-30 cm of before RW. Numbers within brackets denote the standard deviation.

| Parameter | Unit | Before RW | | | After RW |
|---|---|---|---|---|---|
| | | 0-10 cm | 10-20 cm | 20-30 cm | 0-10 cm |
| BD | kg dw l$^{-1}$ | 0.44 (0.14) | 0.35 (0.19 | 0.20 (0.06) | 0.24 (0.06) |
| OM | % | 34 (13) | 50 (27) | 70 (19) | 61 (13) |
| OM | g dw l$^{-1}$ | 135 (18) | 138 (23) | 132 (14) | 138 (11) |
| Total-P | mmol l$^{-1}$ | 15 (3.4) | 7.1 (3.2) | 3.2 (1.5) | 5.3 (2.0) |
| Olsen-P | mmol l$^{-1}$ | 0.98 (0.15) | 0.51 (0.22) | 0.18 (0.11) | 0.30 (0.13) |
| Total-Fe | mmol l$^{-1}$ | 173 (73) | 167 (117) | 53 (29) | 61 (27) |
| Total-S | mmol l$^{-1}$ | 48 (15) | 61 (18) | 73 (13) | 87 (14) |

Surface water and pore water chemistry was measured during the whole measurement period in 2020 (Figure 5 and Figure 6). Surface water ammonium ($NH_4^+$) and nitrate ($NO_3^-$) concentrations were relatively low throughout the growing season (< 25 µmol l$^{-1}$ and < 8 µmol l$^{-1}$, respectively). The concentration of phosphate ($PO_4^{3-}$) was also low throughout the growing season in both *Typha* compartments (< 2.5 µmol l$^{-1}$) but increased in the inlet water ditch to 9.2 µmol l$^{-1}$ in July. The pH varied between 7 and 8.6, with the highest values in the period May-September and the lowest in winter (Table A1, Appendix). The Chloride ($Cl^-$) concentration also showed a clear seasonal pattern, with relatively low concentrations in winter (~20 mmol l$^{-1}$ in February) and high concentrations in summer (~60 mmol l$^{-1}$ in August 2020). Total sulphur (TS) was highly variable over time but was generally lower in both *Typha* compartments (221-887 µmol l$^{-1}$ in *T. Angustifolia* and 127-432 µmol l$^{-1}$ in *T. latifolia*), compared to the *Azolla* compartment (387-1291 µmol l$^{-1}$) and the water inlet ditch (495-1414 µmol l$^{-1}$).

In the pore water, $NH_4^+$ and total phosphorus (TP) concentrations were low throughout the year in both *Typha* compartments (< 30 µmol l$^{-1}$ and < 4.4 µmol l$^{-1}$, respectively). In the *Azolla* compartment, however, $NH_4^+$ was substantially higher with concentrations ranging from 130 µmol l$^{-1}$ in July 2020, to 650 µmol l$^{-1}$ in September 2020. TP increased from 10 µmol l$^{-1}$ in March to 50 µmol l$^{-1}$ in October. The $Cl^-$ concentration in the pore water showed a seasonal pattern as well, with 20-40 mmol l$^{-1}$ in March 2020 to 45-70 mmol l$^{-1}$ in October 2020. In the *Azolla* compartment, pore water was very iron (TFe)- and sulphur (TS)-rich in March and April 2020 (> 3000 and > 1000 µmol l$^{-1}$, respectively, Table A2, Appendix), but dropped to concentrations similar to both *Typha* compartments during the summer. Sulphide ($S^{2-}$) concentrations in the pore water were very low (< 0.2 µmol l$^{-1}$) throughout the year in all three compartments (Table A2, Appendix).

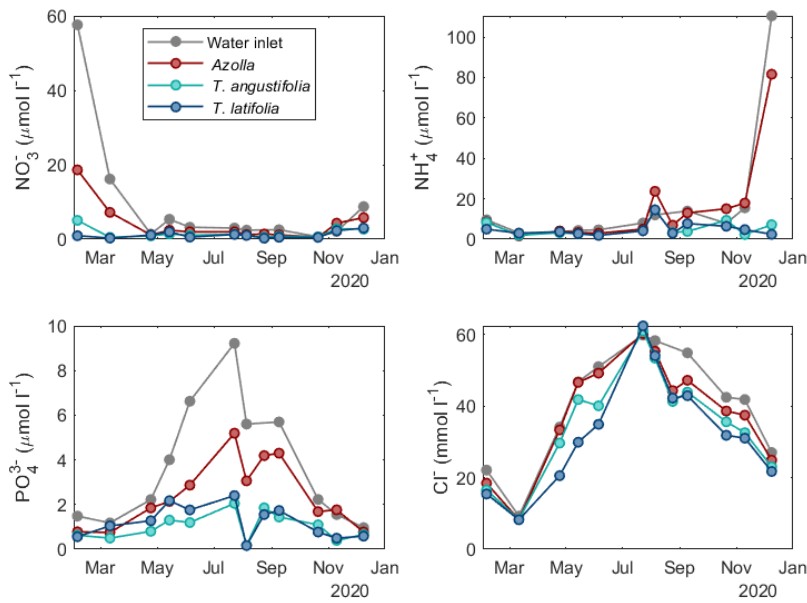

**Figure 5 Surface water chemistry of nitrate ($NO_3^-$), ammonium ($NH_4^+$), phosphate ($PO_4^{3-}$), and chloride ($Cl^-$) measured in the different compartments of the three paludicrops, and in the water inlet ditch at different moments in time.**

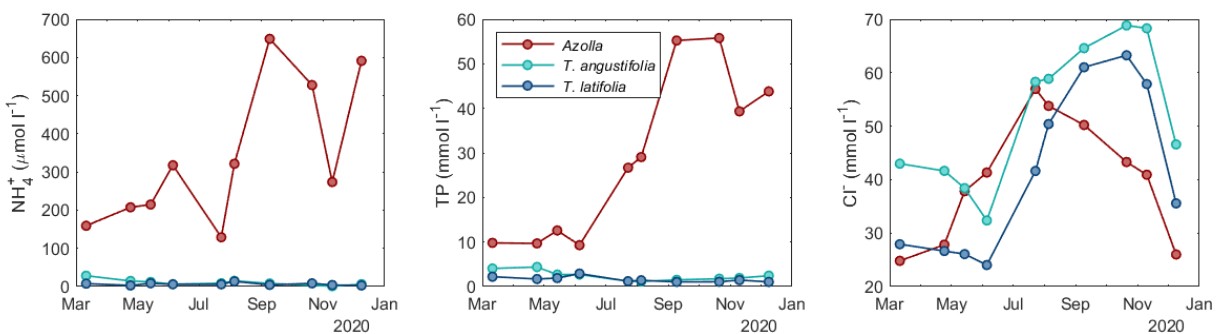

**Figure 6 Pore water chemistry of ammonium ($NH_4^+$), total phosphorus (TP), and chloride ($Cl^-$) measured in the soil of the different compartments of the three paludicrops at different moments in time.**

### 3.3 Methane fluxes

Diffusive $CH_4$ fluxes were measured with chambers and ebullition with bubble traps for all three paludicrops. For interpolating

$CH_4$ fluxes to come to a yearly budget, relations with environmental variables were analysed for the three paludicrops. Diffusive $CH_4$ fluxes from *T. latifolia* and *T. angustifolia* showed the strongest correlation with living above ground biomass ($R^2 = 0.78$ and 0.63, respectively). However, this variable is not useful for interpolation, since we do not have biomass data between the measurement campaigns. Soil temperature was the second-best explanatory factor for $CH_4$ emission for both *Typha* species ($R^2 = 0.63$ for *T. latifolia*; $R^2 = 0.58$ for *T. angustifolia*), and had a very strong correlation with aboveground

biomass ($R^2$ = 0.94 for *T. latifolia*; $R^2$ = 0.95 for *T. angustifolia*). For *Azolla*, water temperature correlated slightly better ($R^2$ = 0.45) than soil temperature ($R^2$ = 0.37). Therefore, for *Typha*, we used soil temperature and for *Azolla* water temperature for interpolating the diffusive $CH_4$ fluxes as showed in Fig. 7B (for a more detailed description of the interpolation, see methods section 2.4).

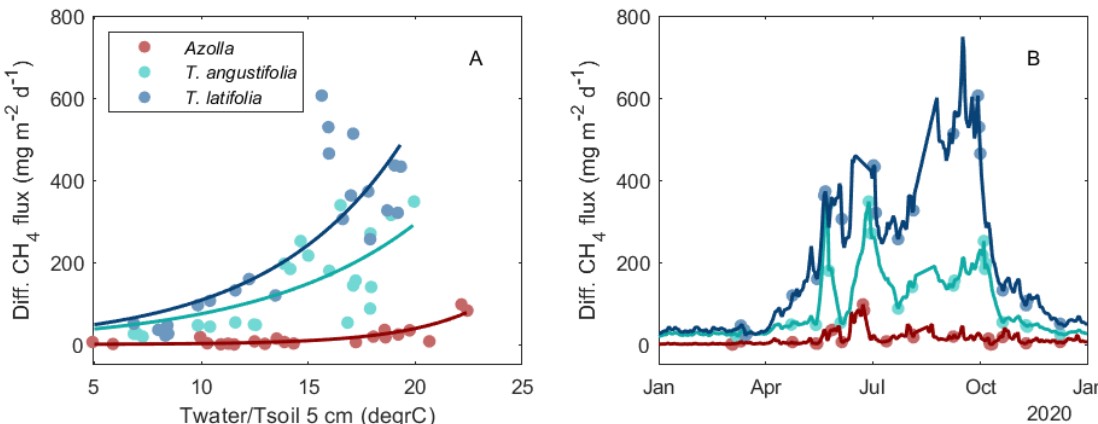

**Figure 7 Relation between daily mean diffusive $CH_4$ flux with water temperature (*Azolla*) or soil temperature at 5 cm depth (*Typha angustifolia*, *Typha latifolia*) (A). Measured (dots) and interpolated (lines) diffusive $CH_4$ flux by using the temperature relation (B).**

Ebullition measurements were considered as the average ebullitive flux between the sampling moments. This created a continuous data series, except for January-February where there were no measurements yet. Ebullitive fluxes for January-
February were assumed to be the same as for March.

The yearly sum of $CH_4$ flux was highest for *T. latifolia* and lowest for *Azolla*, with the highest absolute and relative contribution of ebullition for *Azolla* (Table 2).

**Table 2 Total $CH_4$ flux for 2020 for *T. angustifolia*, *T. latifolia* and *Azolla*. Total flux consists of diffusive flux and ebullition flux. The**
**number between brackets denotes the standard deviation, representing the variation between the measurement replicates. For the calculation of $CH_4$ fluxes in $CO_2$ equivalent ($CO_2$-eq), a GWP$_{100}$ of 27.2 is used (IPCC, 2021).**

| Species | CH$_4$ diffusion (g CH$_4$ m$^{-2}$ yr$^{-1}$) | CH$_4$ ebullition (g CH$_4$ m$^{-2}$ yr$^{-1}$) | Total CH$_4$ flux (g CH$_4$ m$^{-2}$ yr$^{-1}$) | Total in CO$_2$-eq (t CO$_2$-eq ha$^{-1}$ yr$^{-1}$) | % ebullition (%) |
|---|---|---|---|---|---|
| *T. angustifolia* | 33.6 (19.2) | 3.2 (5.2) | 36.9 (20.0) | 10.0 | 9 |
| *T. latifolia* | 76.2 (42.7) | 8.5 (12.2) | 84.8 (49.3) | 23.1 | 10 |
| *Azolla* | 5.1 (5.9) | 17.2 (24.2) | 22.3 (25.9) | 6.1 | 77 |

### 3.4 Carbon dioxide fluxes

$CO_2$ fluxes always reflect a combination of different processes: daytime uptake of $CO_2$ by plant photosynthesis (GPP), and ecosystem respiration ($R_{eco}$) as the sum of plant respiration for maintenance and growth (autotrophic respiration) and soil respiration (heterotrophic respiration). $R_{eco}$ showed large differences between the different paludicrops and over the seasons (Figure 8). For the three different species, the total year sum of $CO_2$ was the highest for *T. latifolia*, but $R_{eco}$ was still around half of that from the reference site (Table 3). *T. latifolia* had a higher GPP compared to *T. angustifolia*, but this could only partly explain the difference in $R_{eco}$. *Azolla* clearly had the lowest GPP (and thus biomass production) and the lowest $R_{eco}$.

However, in relation to the GPP, $R_{eco}$ was relatively high, resulting in the lowest net uptake of $CO_2$ (NEE) (Table 3).

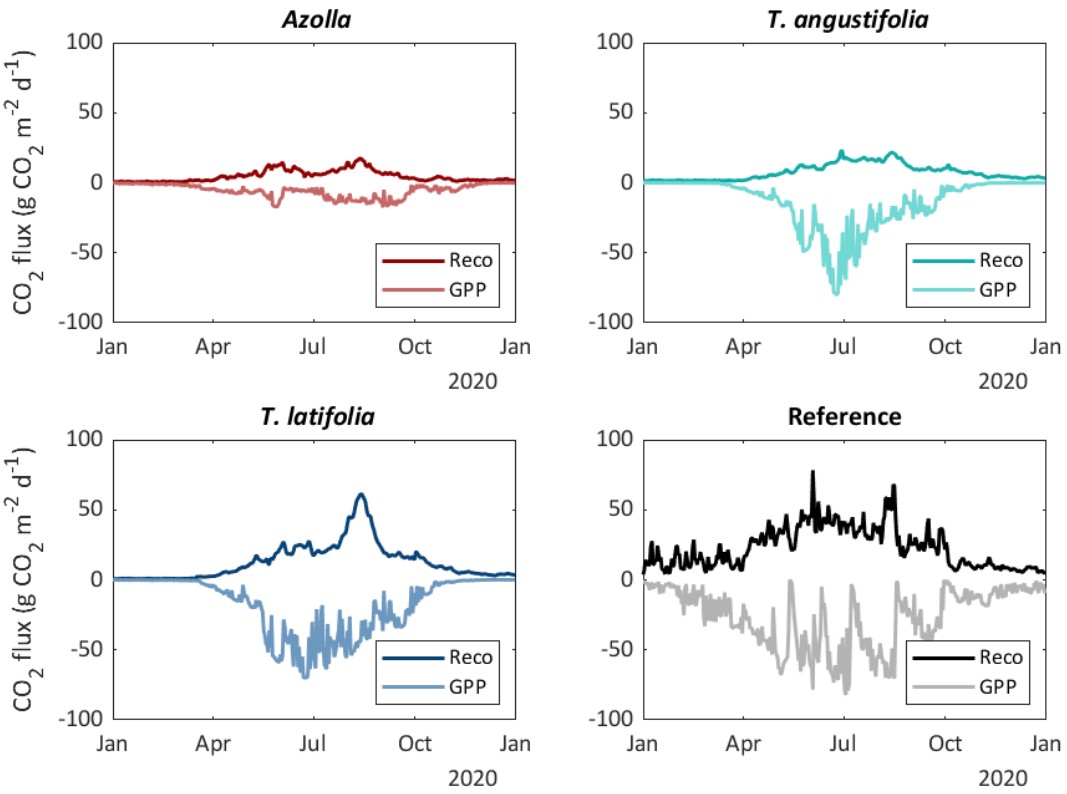

**Figure 8 Estimated daily average ecosystem respiration ($R_{eco}$) and gross primary production (GPP) for the three paludicrops and the drained reference site.**

To derive an annual $CO_2$ balance, the fluxes were interpolated (see methods Sect. 2.4) over the entire year (Figure 8). To be conservative with the $CO_2$ balance, it is assumed that all harvested biomass will be decomposed at some point and this was therefore converted to $CO_2$ (as C-export term) and added to the NEE to come to the complete $CO_2$ balance. Biomass could, however, be stored sustainably in for instance building material, which would reduce the total $CO_2$ flux. On the other hand, if

biomass is used for fodder, carbon could be released as $CH_4$ again, increasing the total greenhouse gas flux. This is not considered within our balances. Harvested biomass for *T. latifolia* and *T. angustifolia* at the end of 2020, was estimated to be 8 and 11 t dm ha$^{-1}$ yr$^{-1}$, respectively. The harvested biomass was corrected for the biomass that was left in the previous year (not harvested), to avoid double counting. In total, we observed that the net uptake of $CO_2$ was greater than the yield, meaning that the $CO_2$ balance results in a $CO_2$ uptake of the system for all crops, with the highest uptake (-1.26 kg $CO_2$ m$^{-2}$ yr$^{-1}$) for *T. latifolia* and lowest uptake for *Azolla* (-0.13 kg $CO_2$ m$^{-2}$ yr$^{-1}$) (Table 3).

**Table 3 Yearly interpolated $CO_2$ fluxes, consisting of ecosystem respiration ($R_{eco}$), gross primary production (GPP), net ecosystem exchange (NEE), carbon removed from harvest (C-export), and the sum of NEE and C-export (Total $CO_2$). The number between brackets denotes the standard deviation, representing the variation between the measurement replicates. For *Typha* C-export two samples were taken, so no standard deviation could be determined.**

| Species | $R_{eco}$ kg $CO_2$ m$^{-2}$ yr$^{-1}$ | GPP kg $CO_2$ m$^{-2}$ yr$^{-1}$ | NEE kg $CO_2$ m$^{-2}$ yr$^{-1}$ | C-export kg $CO_2$ m$^{-2}$ yr$^{-1}$ | Total $CO_2$ kg $CO_2$ m$^{-2}$ yr$^{-1}$ |
|---|---|---|---|---|---|
| *T. angusifolia* | 2.78 | -5.15 | -2.37 (1.9) | 1.23 (0.89) | -1.14 (2.1) |
| *T. latifolia* | 4.72 | -6.45 | -1.72 (1.5) | 0.46 (0.15) | -1.26 (1.6) |
| *Azolla* | 1.73 | -2.04 | -0.31 (0.43) | - | -0.13 (0.43) |
| Reference | 8.38 | -9.79 | -1.41 (0.12) | 3.47 (0.39) | 2.06 (0.41) |

### 3.5 Total greenhouse gas balance

For the greenhouse gas (GHG) balance, both $CO_2$ and $CH_4$ emissions in $CO_2$-equivalents ($CO_2$-eq; $GWP_{100}$ of 27.2, IPCC, 2021) were summed up. For the reference site, we did not measure $CH_4$ fluxes in 2020, however, $CH_4$ fluxes were assumed to be zero based on $CH_4$ flux measurements on the same site in 2019 (Gremmen et al., 2022).

For the paludicrops, only *T. angustifolia* had a higher uptake of $CO_2$ (also considering C-export) than the $CH_4$ in $CO_2$-eq that was emitted, making it a net GHG sink (-1.4 t $CO_2$-eq ha$^{-1}$ yr$^{-1}$). With the other two species, $CH_4$ emission was higher than $CO_2$ uptake, with a higher emission for *T. latifolia* (10.5 t $CO_2$-eq ha$^{-1}$ yr$^{-1}$) than for *Azolla* (2.9 t $CO_2$-eq ha$^{-1}$ yr$^{-1}$) (Figure 9). Nevertheless, all paludicrops had a lower net GHG emission than the reference site (20.6 t $CO_2$-eq ha$^{-1}$ yr$^{-1}$). However, in this balance the potential $CO_2$ emission from the topsoil removal from the paludiculture site (557 t $CO_2$ ha$^{-1}$) is not accounted for.

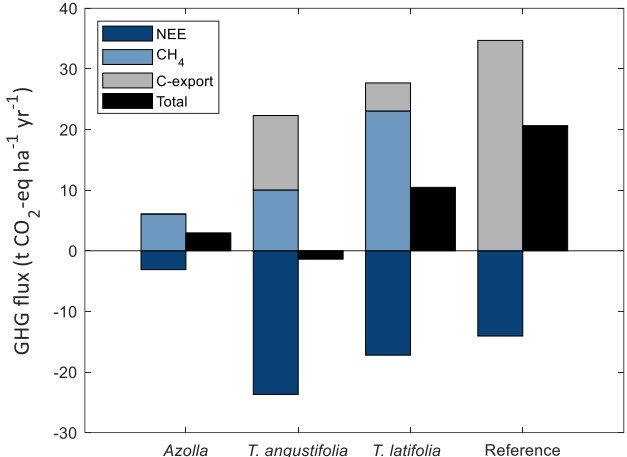

**Figure 9 Greenhouse gas (GHG) balance for the three paludicrops and the reference site. GHG balance consists of net ecosystem exchange of $CO_2$ (NEE), carbon removed by harvest (C-export) and $CH_4$ flux (consisting of ebullition and diffusive fluxes) expressed in $CO_2$ equivalent (GWP$_{100}$ of 27.2, IPCC, 2021), and the total net flux as the sum of the three terms. *Typha* yield was corrected for the biomass that was left in 2019. Therefore, it does not represent the potential yield from the *Typha* fields.**

Since $CH_4$ has a relatively short lifetime in the atmosphere (it reacts with hydroxyl radicals to form $CO_2$ and water vapor), the contribution of $CH_4$ to the radiative forcing of our planet has a different behaviour over time than $CO_2$. This creates a complex trade-off between reducing $CO_2$ emission from drained peatlands vs rewetting and creating $CH_4$ emission on the long-term (Günther et al., 2020). To visualise this, we used the radiative forcing model of Günther et al. to see what the long-term effect is of rewetting and topsoil removal for paludiculture compared to the reference site for $CO_2$ and $CH_4$. In this case we assumed that the current measured $CO_2$ and $CH_4$ fluxes will continue until the end of the century, and that the removed carbon from the topsoil will be decomposed to $CO_2$ within 27 years (assuming the $CO_2$ emission will be the same as from the reference site). We also assumed that the harvested biomass (yield) is decomposed again to $CO_2$ within that same year. The total radiative forcing is the sum of the contribution from $CH_4$ emission, $CO_2$ from topsoil removal, and net $CO_2$ flux (NEE + yield) and is calculated for an area of 220,000 ha (area of arable drained peatlands in the Netherlands). The results show that in the first decades the total radiative forcing for all paludicrops is higher than for the reference site, due to the combination of increasing radiative forcing from $CH_4$ emission and $CO_2$ emission from topsoil removal (Figure 10). After 27 years, the emission from topsoil removal stops and radiative forcing from $CH_4$ emission steadily flattens off. From the year 2063, radiative forcing for (coincidentally) both *Azolla* and *T. angustifolia* becomes lower than for the reference site. For *T. latifolia* this is not going to happen until the year 2113. Eventually, with the additional flux of topsoil removal, *T. latifolia* has the highest impact on the radiative forcing in the year 2100: $4.2 \cdot 10^{-4}$ W m$^{-2}$ compared to $1.9 \cdot 10^{-4}$ W m$^{-2}$ for *Azolla*, $1.3 \cdot 10^{-4}$ W m$^{-2}$ for *T. angustifolia*, and $3.6 \cdot 10^{-4}$ W m$^{-2}$ from the reference site. Topsoil removal contributes to that with about $1.0 \cdot 10^{-4}$ W m$^{-2}$ for the three paludicrops.

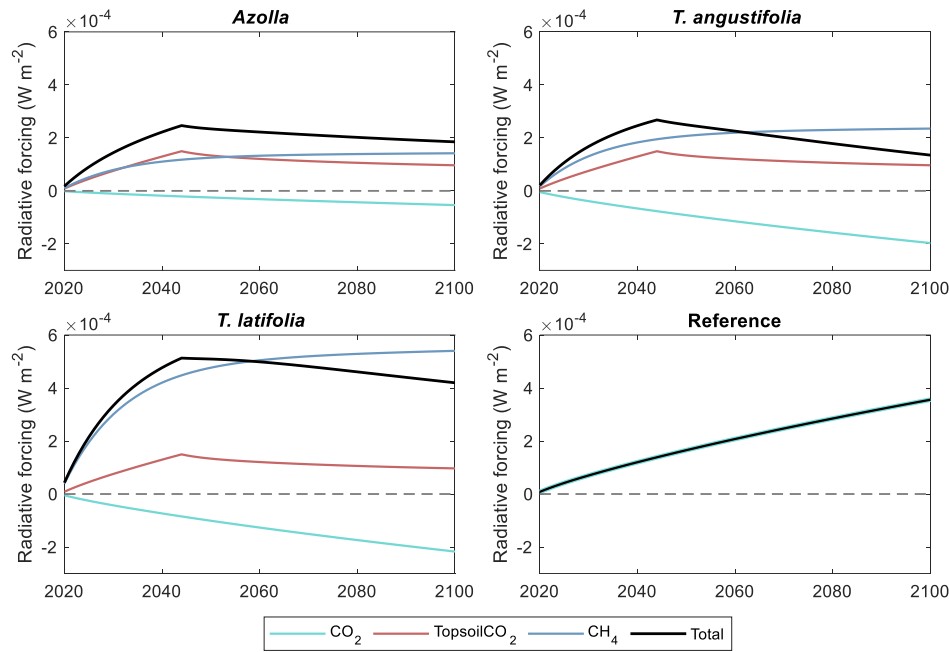

**445**

**Figure 10 Contribution of paludicrops and reference site to radiative forcing for CO₂ flux (NEE + yield), estimated CO₂ emission from topsoil removal, and CH₄ emission. The topsoil is assumed to be completely decomposed within 27 years. The radiative forcing is calculated for a land surface area of 220,000 ha, which is the area of drained arable peatlands in the Netherlands.**

## 4. Discussion

**450** **4.1 Differences in CH₄ flux of the three paludicrops**

We found large differences in diffusive and ebullitive CH₄ fluxes from the three paludicrop species. The differences in CH₄ fluxes, with lowest diffusive and highest ebullitive fluxes for *Azolla*, can be well explained by the differences in growth forms and species-specific characteristics. A more thorough discussion of the effects of these plants on CH₄ emissions can be found in Vroom et al. (Under review). Briefly, *Azolla,* a free-floating plant without roots in the soil, neither releases root exudates to

**455** the sediment, nor transports sediment CH₄ to the atmosphere. Moreover, radial oxygen loss (ROL) from roots can lead to oxidation of up to 70% of the produced CH₄ (Kosten et al., 2016). This may explain the relatively low CH₄ diffusion from *Azolla*, which has been found for other free-floating species before (Attermeyer et al., 2016). On the other hand, CH₄ emissions by ebullition are still substantial, probably due to the release of dead roots and root exudates to the water, providing carbon for methane production. In the case of *Typha*, plant mediated transport is substantial (Bendix et al., 1994; Yavitt and Knapp, 1998;

**460** White and Ganf, 2000) and CH₄ production in the sediment can be increased by the supply of carbon through the roots as root exudates, which is an important source for CH₄ production (Bastviken et al., 2023). The plant transport of CH₄ causes the CH₄ concentration in the soil to decrease, which leads to a lower ebullition flux (Van der Nat et al., 1998; Grünfeld and Brix, 1999; Van den Berg et al., 2020). So, although rates of methane production may be higher due to the supply of easily degradable

carbon, which will increase over the course of the season as the plants grow larger (Joabsson and Christensen, 2001), the build-
465 up of $CH_4$ in the sediment will remain low. Lower emissions from *T. angustifolia* than *T. latifolia* may be explained by the greater ability of *T. angustifolia* to build up pressure in the stem and higher ROL rates compared to *T. latifolia* (Bendix et al., 1994; Matsui Inoue and Tsuchiya, 2008), resulting in higher rates of methane oxidation (Bendix et al., 1994). Additionally, in 2020 90% of *T. latifolia* had been damaged by the Webb's Wainscot and/or the Bulrush Wainscot, which may have reduced pressurized flow (Armstrong et al., 1996).

The total $CH_4$ emission in 2020 for *T. latifolia* was high, both relative to the other species and in absolute terms (84.8 g $CH_4$ $m^{-2}$ $yr^{-1}$). Emissions were around a factor ~1.7 higher than what was found in a similar experiment in the Netherlands (Buzacott et al., Under review), but the same magnitude was found in a boreal lake in Canada (Desrosiers et al., 2022). However, Rey-Sanchez et al. (2018) show even higher emissions from a natural system in the USA than what we found (292 g $CH_4$ $m^{-2}$ $yr^{-1}$). They hypothesize that the high fluxes could be attributed to high DOC input. This could also be the case in our site, as high
TOC values (5.3 mmol $l^{-1}$, Table A1) were found in the inflow ditch water. Another reason could be plant stress factors, like salinity level and herbivory, resulting in enhanced die-off of plant material. Both could lead to higher substrate availability for $CH_4$ production. *T. angustifolia* was less affected by herbivory and has a higher salt tolerance than *T. latifolia* (McMillan, 1959).

The lower $CH_4$ emission for *T. angustifolia* compared to other rooting wetland plants was also found in another study, where
fluxes from *Phalaris* and *Phragmites* were a factor 1.7 and a factor 2 higher, respectively (Maltais-Landry et al., 2009). Absolute $CH_4$ emissions from *T. angustifolia* vary strongly from ~11 g $CH_4$ $m^{-2}$ $yr^{-1}$ in a constructed wetland (Maltais-Landry et al., 2009), to ~176 g $CH_4$ $m^{-2}$ $yr^{-1}$ in a natural wetland in Canada (Strachan et al., 2015), compared to our 36.9 g $CH_4$ $m^{-2}$ $yr^{-1}$. But not many studies were found.

For *Azolla*, all studies on $CH_4$ fluxes we found were conducted in combination with rice growth. These studies show in general
a decrease in $CH_4$ emission with the addition of *Azolla* to rice paddies (Bharati et al., 2000; Liu et al., 2017; Xu et al., 2017; Kimani et al., 2018). This is in line with our observations, where we found a factor 1.7-3.8 lower emissions from *Azolla* compared to *Typha*.

Overall, our measured $CH_4$ fluxes were high despite the topsoil removal and the brackish conditions, which were expected to reduce $CH_4$ production, due to the removal of easily degradable carbon (Harpenslager et al., 2015; Quadra et al., 2023) and
490 the reducing effect of salinity on $CH_4$ production (Van der Gon and Neue, 1995; Minick et al., 2019), respectively.

**4.2 Greenhouse gas balance**

Paludiculture reduced the net $CO_2$ balance (including C-export) to a large extent compared to the reference site, going from a net source (+20 t $CO_2$ $ha^{-1}$ $yr^{-1}$) to a net sink (-1.3 t $CO_2$ $ha^{-1}$ $yr^{-1}$ *Azolla*; -11.4 t $CO_2$ $ha^{-1}$ $yr^{-1}$ *T. angustifolia*; and -12.6 t $CO_2$ $ha^{-1}$ $yr^{-1}$ *T. latifolia*). This shows that the rewetting was effective to stop peat oxidation. Adding the $CH_4$ emission in $CO_2$-eq.
also resulted in a lower GHG balance for the paludicrops compared to the reference site. It is uncertain to what extent the carbon storage measured in the *Typha* species will continue in the future. As the species were introduced in the years before

(2018-2019), it is likely that there is still biomass build-up, like rhizomes, which will come to a steady state and reduces the net uptake. In literature, lower uptake values for *Typha* were found, like the range of NEE of -4 to +5 t $CO_2$ ha$^{-1}$ yr$^{-1}$ from a rewetted fen in Belarus (Minke et al., 2016), compared to our -24 and -17 t $CO_2$ ha$^{-1}$ yr$^{-1}$ for *T. angustifolia* and *T. latifolia*, respectively. With *Azolla* the biomass completely disappeared by the end of the measurement year (die-off due to herbivory), and the net $CO_2$ emission was close to zero. From the growth rate observed at the same field site one year later the expected biomass in a growing season could potentially reach 23-35 t dm ha$^{-1}$ yr$^{-1}$ (Gremmen et al., 2022). For *Typha*, herbivory occurred more in *T. latifolia*, resulting in higher biomass die-off, which most likely caused more respiration than in *T. angustifolia*.

The C-export term in *Typha* contributes to the net $CO_2$ balance with about 27-51 % (Table 3). This term is, however, not the C from the total produced biomass, since dead biomass from the previous year was subtracted to make the balance right. Yields of 10-25 t dm ha$^{-1}$ yr$^{-1}$ can be possible (Geurts and Fritz, 2018), while our site showed 8 (*T. latifolia*) to 11 (*T. angustifolia*) t dm ha$^{-1}$ yr$^{-1}$.

The yield term assumes that all harvested carbon is decomposed to $CO_2$ again. This is the case if biomass is burned or used as fodder (although C-emission can also be in the form of $CH_4$ in this case), but if biomass is used sustainably for long term storage such as building material, this C-export should not be accounted for in the carbon/GHG balance. However, to know the exact GHG effect of biomass storage, a life cycle assessment is needed to account for all other emission terms. De Jong et al., (2021) estimated that using *Typha* as insulation material, emissions from cultivating and processing *Typha* would be 9.7 t $CO_2$-ha$^{-1}$ yr$^{-1}$, which almost compensates the biomass harvest in *T. latifolia*. They also conclude that the largest GHG gain is reducing the peat oxidation and not in the biomass use.

When taking into account the impact of topsoil removal on the carbon balance, the perspective changes substantially. If all the carbon that is removed from the top 20 cm (15.8 kg m$^{-2}$) is not stored under anoxic conditions, an amount of 557 t $CO_2$ ha$^{-1}$ will be released over the period needed to decompose that carbon. That is the same amount the reference site is emitting in 27 years. With the radiative forcing calculation for the different paludicrops (Figure 10), topsoil removal is the largest contributor for *T. angustifolia* and around half of the contribution for *Azolla* by 2100. Therefore, if topsoil removal is applied, one should consider the potentially large $CO_2$ emission associated with topsoil decomposition. Additionally, Quadra et al. (2023) show that only 5 cm of topsoil removal might be sufficient to significantly reduce $CH_4$ emissions, suggesting that careful consideration of the amount of topsoil removal is warranted.

The impact of $CO_2$ and $CH_4$ terms on radiative forcing (global warming), are not behaving the same over time. With continuous emission, $CH_4$ in the atmosphere will reach equilibrium so that the contribution to radiative forcing stabilises, while continuous $CO_2$ emission causes an always increasing radiative forcing. The $CO_2$ emission from topsoil removal will only last until all the organic carbon is decomposed. We visualized this in Figure 10, with pointing out the differences among the paludicrops and reference site. In the first ~40 years, the reference site has a lower impact on climate warming than the paludicrops *Azolla* and *T. angustifolia*, for *T. latifolia* this will take 50 years longer. On the long term, however, it is better to rewet to prevent $CO_2$ emission, since the impact of $CH_4$ emission will reduce over time. This only applies if the GHG emission will not change over time. This is, however, a very uncertain assumption, since we only measured one year just after rewetting. As discussed before,

it is likely that the net $CO_2$ uptake by the *Typha* species reduce if above- and below-ground biomass production is stabilized. Other factors that influence $CH_4$ emission (water table, DOC inflow) could also change over time.

A missing term in the GHG balance is $N_2O$ emission, which can be a significant term in drained peatlands. IPCC's emission factor for drained peatlands with an N fertilizer application of 250 kg N ha$^{-1}$ yr$^{-1}$ is 19 kg $N_2O$ ha$^{-1}$ yr$^{-1}$ (Liang and Noble, 2019), which is equal to 5.2 t $CO_2$-eq ha$^{-1}$ yr$^{-1}$ (GWP$_{100}$ of 273; IPCC, 2021). Production of $N_2O$ increases with increasing water table, but in complete anaerobic conditions $N_2O$ is reduced to $N_2$ (Weier et al., 1993). Therefore, the $N_2O$ emissions from the paludicrops are expected to be close to zero, but the GHG balance from the reference site is expected to increase from 20.6 to around 25 t $CO_2$-eq ha$^{-1}$ yr$^{-1}$.

## 4.3 Biochemical interactions with paludicrops

Our results indicate that both *Typha* spp. effectively reduced nutrients (N+P) in the surface water and pore water, which is consistent with the results of Vroom et al. (2018). In the *Azolla* compartment N (as ammonium) and P accumulated in the pore water, but concentrations in the surface water were also low. The removal of ~20 cm of the topsoil, and with that a reduction of approximately 65% of soil total-P, probably resulted in strong P-limiting conditions for *Azolla* (Temmink et al., 2018) and possibly also for both *Typha* spp. (Lorenzen et al., 2001).

Rewetting without topsoil removal probably has a positive effect on biomass production, especially for *Azolla*. Recent studies have shown, however, that this could lead to high $CH_4$ emissions (Harpenslager et al., 2015; Quadra et al., 2023). Our results indicate that smart crop-choices can, to an extent, mitigate these effects. The high phosphate mobilisation often associated with rewetting of former agricultural drained peatland without topsoil removal could create the right conditions for *Azolla* cultivation, while also reducing $CH_4$ emissions compared to *Typha* cultivation. *Azolla* can be used as a temporary crop while the phosphorus mobilisation-rates after rewetting are high (Forni et al., 2001; Temmink et al., 2018) and the system adjusts to continuously waterlogged conditions. Once the phosphorus-flux to the overlying water is reduced other (rooting) paludicrops could be introduced.

In coastal areas salinity plays an important role in crop choice. *T. angustifolia* is more salt-tolerant than *T. latifolia*. The upper limit for *T. latifolia* lies between 1.6-2.7 g l$^{-1}$ (Anderson, 1977), which is similar to concentrations we observed (2.2 g l$^{-1}$) and may partly explain the inhibited growth. For *T. angustifolia* our measured concentrations were lower than the upper limit of 7.2-8.8 g l$^{-1}$ (Sinicrope et al., 1990).

Even though *T. latifolia* showed higher $CH_4$ emissions and lower salt tolerance compared to *T. angustifolia*, there are also advantages to use this species as a paludicrop. *T. latifolia* is considered to be more suitable for building materials due to the higher yield and more optimal diameter (Haldan et al., 2022). Furthermore, *T. latifolia* can also grow better at lower water tables, and $CH_4$ emissions could be significantly reduced if the water table drops below soil surface (Haldan et al., 2022). The effect of water table on $CH_4$ emissions was also studied within this experimental set-up and results can be found in Vroom et al. (Under review).

**Conclusion**

Our results show that all paludiculture crops reduce GHG emission compared to an intensively used drained fen grassland, with highest reduction for *Typha angustifolia* and lowest for *Typha latifolia*. $CH_4$ emission in $CO_2$-eq is very variable per species but can be as high or higher than the $CO_2$ emission from drained peatland but is (partly) compensated by net $CO_2$ uptake.

*Typha* is a rooting plant, resulting in plant mediated gas transport from the sediment to the atmosphere and easily degradable carbon input into the sediment. This leads to higher total $CH_4$ emission compared to *Azolla*, but also to a lower contribution of ebullition to the total $CH_4$ flux.

Topsoil removal did not lead to low $CH_4$ emissions, especially not for *Typha latifolia*. What did change was nutrient availability with topsoil removal, probably leading to limiting growth of all species, in particular for *Azolla*. An undesired effect of topsoil removal is the potentially high $CO_2$ emission from the removed soil carbon. This term can contribute significantly to the total radiative forcing caused by the paludicrops.

In our case study *Azolla* and *T. angustifolia* seem to have a high potential for peatland rewetting to reduce the impact on climate change. A follow-up study without e.g. topsoil removal would be interesting to see if *Azolla* would be more productive while keeping $CH_4$ emissions low. Also a longer measurement period and a study without herbivory would be useful to come to more robust GHG balances.

**Data availability**

All raw data can be provided by the corresponding authors upon reasonable request.

**Author contribution**

BvdR, MvdB and AS designed the experiment. YvdV was responsible for the research set-up at the reference site. TG, RV, JvH, JB and CvH carried the experiment out. TG and RV did most of the data preparation. MvdB and TG prepared the manuscript. MvdB did most of the writing, followed by TG and RV. All other authors contributed to editing the manuscript.

**Competing interests**

The contact author has declared that neither of the authors has any competing interests.

## Acknowledgement

The measurements on the paludiculture site were part of the Peat Innovation Program initiated by the Association for Agricultural Nature and Landscape Management; Water, Land & Dijken (WLD) and the landscape conservation organisation
Landschap Noord-Holland (LNH). The measurements on the reference site were part of the Dutch national research program NOBV funded by the Dutch ministry of Agriculture, Nature and Food Quality. RV and BvdR were supported by the NWO-TTW-project AZOPRO (Project no. 16294).

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

  **Appendix A** Biogeochemical data

**Table A1 Surface water chemistry of pH, alkalinity (Alk), electric conductivity (EC), total inorganic carbon (TIC), nitrate ($NO_3^-$), ammonium ($NH_4^+$), phosphate ($PO_4^{3-}$), chloride (Cl), total iron (TFe), potassium (K), magnesium (Mg), sodium (Na), total phosphorus (TP), total sulphur (TS), total organic carbon (TOC) and total organic nitrogen (TON) measured in the plots of the three different paludicrops. Measurements were taken over the whole season (n = 9-12) and averaged. The numbers between the brackets denote the minimum and maximum measured.**

| Variable | Unit | Inlet water | *Azolla* | *T. angustifolia* | *T. latifolia* |
|---|---|---|---|---|---|
| pH | n.a. | 7.8 [7.3, 8.4] | 7.6[7.0, 8.4] | 7.4 [7.0, 7.7] | 7.7 [7.1, 8.6] |
| Alk | meq l−1 | 5.2 [2.1, 7.2] | 4.9 [1.8, 6.8] | 4.4 [1.8, 6.5] | 4.1 [1.8, 5.8] |
| EC | mS cm−1 | 4.8 [1.4, 7.3] | 4.4 [1.3, 6.7] | 4.0 [1.3, 6.2] | 3.6 [1.2, 6.3] |
| TIC | mmol l-1 | 5.0 [1.7, 7.6] | 5.0 [1.6, 7.3] | 4.6 [1.4, 6.3] | 4.1 [1.4, 6.0] |
| $NO_3^-$ | µmol l−1 | 9.4 [0.65, 58] | 4.0 [0.55, 19] | 1.6 [0.50, 5.1] | 1.1 [0.28, 3.0] |
| $NH_4^+$ | µmol l−1 | 18 [3.1, 110] | 15 [1.7, 82] | 5.2 [1.9, 14] | 4.9 [1.9, 15] |
| $PO_4^{3-}$ | µmol l−1 | 3.7 [0.97, 9.2] | 2.4 [0.75, 5.2] | 1.0 [0.17, 2.1] | 1.2 [0.16, 2.4] |
| $Cl^-$ | mmol l-1 | 41 [9.2, 60] | 39 [8.2, 60] | 36 [8.3, 61] | 33 [8.2, 62] |
| TFe | µmol l−1 | 39 [30, 72] | 32 [17, 49] | 33 [10, 66] | 55 [18, 160] |
| K | mmol l-1 | 1.3 [0.20, 2.8] | 1.2 [0.18, 2.7] | 1.1 [0.18, 2.0] | 1.0 [0.18, 2.0] |
| Mg | mmol l-1 | 3.5 [0.87, 5.1] | 3.2 [0.73, 4.7] | 2.8 [0.70, 4.3] | 2.6 [0.70, 4.3] |
| Na | mmol l-1 | 40 [7.9, 71] | 37 [7.1, 66] | 33 [7.1, 53] | 30 [6.9, 54] |
| TP | µmol l−1 | 9.9 [3.6, 19] | 7.4 [3.0, 14] | 3.6 [2.5, 5.3] | 4.4 [2.2, 6.7] |
| TS | mmol l-1 | 1.1 [0.50, 1.4] | 0.78 [0.39, 1.3] | 0.46 [0.22, 0.89] | 0.25 [0.13, 0.43 |
| TOC | mmol l-1 | 5.3 [3.3, 8.6] | 5.1 [3.7, 8.1] | 4.8 [3.6, 6.7] | 5.3 [3.4, 6.8] |
| TON | mmol l-1 | 0.28 [0.20, 0.42] | 0.26 [0.19, 0.36] | 0.21 [0.17, 0.27] | 0.22 [0.17, 0.29] |

**Table A2 Pore water chemistry of pH, electric conductivity (EC), total inorganic carbon (TIC), nitrate ($NO_3^-$), ammonium ($NH_4^+$), chloride (Cl), total iron (TFe), potassium (K), magnesium (Mg), sodium (Na), total phosphorus (TP) and total sulphur (TS) , hydrogen sulphide ($H_2S$), dissolved organic carbon (DOC) and dissolved organic nitrogen (DON) measured in the plots of the three different paludicrops. Measurements were taken over the whole season (n = 8-10) and averaged. The numbers between the brackets denote the minimum and maximum measured.**

| Variable | Unit | *Azolla* | *T. angustifolia* | *T. latifolia* |
|---|---|---|---|---|
| pH | n.a | 6.4 [6.2, 6.7] | 4.8 [4.7, 5.0] | 6.5 [6.3, 6.7] |
| EC | mS cm−1 | 4.6 [3.5, 6.5] | 5.6 [4.0, 7.3] | 4.7 [3.1, 7.0] |
| TIC | mmol l-1 | 8.2 [6.4, 11] | 4.4 [3.1, 5.8] | 7.9 [5.8, 10] |
| $NO_3^-$ | µmol l−1 | 3.1 [0.9, 8.6] | 1.8 [0.4, 4.7] | 1.1 [0.3 2.7] |
| $NH_4^+$ | µmol l−1 | 339 [129, 649] | 11 [1, 28] | 6.4 [2.9, 14] |
| $Cl^-$ | mmol l-1 | 40 [25, 57] | 52 [32, 69] | 41 [24, 63] |
| TFe | µmol l−1 | 1402 [114, 3275] | 411 [298, 514] | 62 [37, 91] |
| K | mmol l-1 | 1.0 [0.28, 1.9] | 1.0 [0.67, 1.4] | 0.95 [0.56, 1.3] |
| Mg | mmol l-1 | 3.2 [2.2, 4.6] | 2.3 [1.3, 3.3] | 3.4 [1.9, 5.5] |
| Na | mmol l-1 | 38 [25, 55] | 48 [32, 64] | 38 [23, 59] |
| TP | µmol l−1 | 29 [9.3, 56] | 2.4 [1.1, 4.4] | 1.6 [1.0, 2.9] |
| TS | mmol l-1 | 0.56 [0.18, 1.3] | 0.27 [0.20, 0.45] | 0.24 [0.11, 0.72] |
| $H_2S$ | µmol l−1 | 0.024 [0.004, 0.072] | 0.052 [0.007, 0.14] | 0.023 [0.005, 0.052] |
| DOC | mmol l-1 | 24 [7.2, 59] | 8.3 [6.3, 11] | 5.2 [4.5, 5.8] |
| DON | mmol l-1 | 1.5 [0.61, 2.8] | 0.34 [0.24, 0.43] | 0.22 [0.19, 0.25] |

**Appendix B** Biomass data

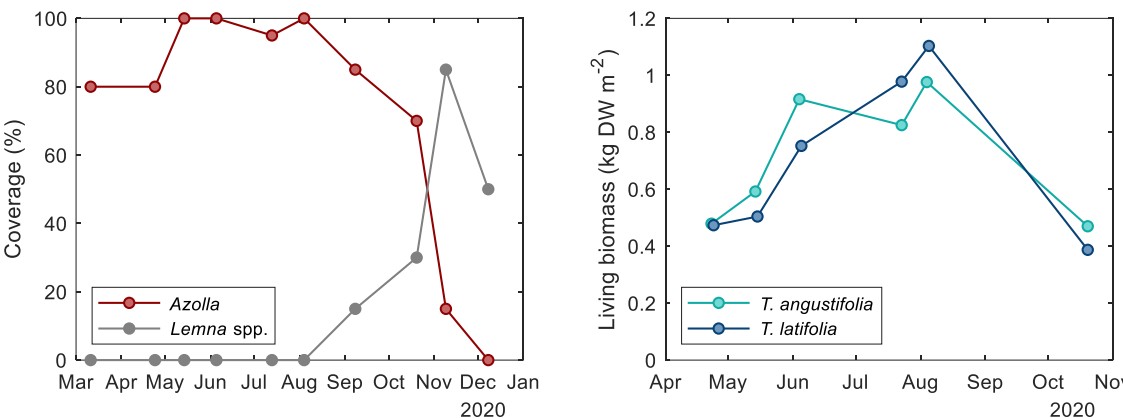

**Figure A1 Coverage of *Azolla* and *Lemna* spp. (left) and estimated living biomass based on stem count and stem height in the chamber plots for *Typha* (right) for measuring year 2020.**