# Peer review of "A case study on topsoil removal and rewetting for paludiculture: effect on biogeochemistry and greenhouse gas emissions from *Typha latifolia*, *Typha angustifolia* and *Azolla filiculoides"

_EGUsphere, 2023_

## Author Comment (AC1)

Anonymous Referee #1, 03 Jan 2024

*We would like to thank Referee #1 for his/her effort to review our paper. We are happy to hear that the reviewer finds the study interesting. We agree with most of the comments and suggestions that are given. In the comments below we will elaborate on how we want to incorporate the suggested changes in our manuscript. Also, if we disagreed with the reviewer, we will elaborate on the argumentation why we do not share his/her vision.*

General: This manuscript describes an interesting case study on the effects of top soil removal, rewetting and the use of different wetland plant species on greenhouse gas emissions. The emissions fluxes were compared between these different plant species, but also with a reference grassland site. It would be nice to mention the total greenhouse gas balance in the abstract also, in CO2 equivalents, so that the reader can directly see the effect of the treatments on the greenhouse gas balance.

*Thank you for the suggestion. We will add the total GHG balances of the treatment and reference site in $CO_2$ equivalents to the abstract.*

Introduction:

L75. This is the first time that I hear that vegetated conditions may have higher CH4 emissions than non-vegetated conditions. Moreover, in the paper of Antonijevic it is stated that the period with elevated CH4 emissions ended with the occurence of cattail. So please correct that reference. And why are there no measurements of non-vegetated conditions in this experiment?

*There are many studies that show that vegetation leads to higher emission (Bodmer et al., 2024; Bastviken et al., 2023; Zhang et al., 2019; Hendriks et al., 2010; Kankaala et al., 2003) with the most important reason the carbon substrate input in the system for methanogens, and plant $CH_4$ transport. However, the oxygen transport to the rootzone also increases $CH_4$ oxidation, which in some cases leads to lower $CH_4$ emission (e.g. Vroom et al. 2018, van der Nat et al., 1998). But indeed, the study of Antonijevic was not the correct reference to back-up our point. We will change the text with including the argumentation and references above.*
*We conducted also $CH_4$ measurements in non-vegetation conditions, but only with the manual chambers, so therefore we did not include it in this paper (but is described in Vroom et al. in review). These data also showed that the treatment without vegetation had the lowest $CH_4$ emission of all treatments.*

L95. No CH4 measurements at the reference site?

*There were $CH_4$ measurements done on the reference site in a different year (2019) with a different chamber system (manual), therefore we did not include the results in this paper. The data are however described in a report. We refer to this report in the discussion.*

Methods: are the two Typha compartments 430 m2 in total, or are they each 430 m2?

*They are 430 $m^2$ each. We will describe this clearer in the methods.*

Figure 1C: indicate the inlet ditch and the water flow.

*In Figure 1 we aim to visualise the general overview of the field site and its geographic location. In Figure 2 the water flow, including inlet and outlet ditch is further described. We do not see added value in including these details in Figure 1.*

L135. Is it realistic to provide only inorganic fertilizer to the reference site? Does this give an underestimation of carbon fluxes to the atmosphere?

*The decision for inorganic fertilizer was made to prevent to have an extra carbon source for the carbon balance. If organic fertilizer was used, carbon input from organic fertilizer should be subtracted from the carbon and GHG balance. However, the carbon content of manure can vary to some extent, and it is unclear how long it exactly takes for the manure to be decomposed again. This results in higher uncertainties in the carbon and GHG balance. This uncertainty is not present with inorganic fertilizer, since there is no extra carbon input which needs to be corrected for.*

L180. It is a weak point that CH4 fluxes have appartently not been measured in the reference site. This flux could be zero of course, but then the authors should mention this. Also no N2O emissions were measured, which could have a major effect on GHG emissions, especially on the reference site. Please discuss the importance of N2O emissions somewhere in the introduction or discussion.

*As mentioned above as well, we will refer to the measured $CH_4$ emission that are described in a report in the discussion. The emissions were very low, hardly contributing to the total GHG emission.*
*$N_2O$ is indeed a missing GHG. With complete inundation of the soil, we do not expect much $N_2O$ from the paludiculture fields. From the reference site, $N_2O$ emission will most likely contribute significantly to the total GHG balance. Therefore, we expect even a larger reduction of GHG emissions from the paludiculture fields. We will mention this in the discussion.*

Results: it would be good to provide the actual biomass harvest values (per m2 or per ha). Now this is only mentioned in the discussion.

*We will add this to the results as well.*

Fig.7 typo (And).

*Will be removed.*

L370. Table 3. Figure 9. Why is all harvested biomass (C-export) considered as CO2 loss and thus as GHG flux? This totally depends on the biomass use. The grass from the reference site will partly be converted in CH4 by cows and the Typha biomass will for example only be converted to CO2 after a long time if it used as building or insulation material. This seems to be an important disclaimer here. The authors mention this in the discussion, but the disclaimer can also be mentioned here already.

*We will add the disclaimer about the use of the biomass already in the results.*

Discussion: how do Typha roots supply easily degradable carbon to the sediment? And is this in a significant order of magnitude to have effects on CH4 production?

*Roots lose carbon by root exudates, which is an easily degradable substrate. This is a very relevant carbon source for $CH_4$ production and thus emission (Bastviken et al. 2023). We will elaborate on this in the discussion.*

L408-410: several typos.

*Typos will be corrected.*

L410: I think that the damage to the T. latifolia plants is een important thing to mention, also in the abstract and conclusions, as it seems to be the reason for the very high methane emissions.

*Herbivory is already mentioned in the abstract and discussion as possible cause for the higher emissions. We will also write it in the conclusions.*

L451-453: the authors mention the CO2 emissions for cultivating and processing Typha here, but do not mention the CO2 (and CH4) emissions for the reference site, i.e. the cultivating and processing of grass, milk, etc. This probably also (more than) compensates for the grass biomass harvest. So please make a fair comparison, or leave the statement about CO2 emissions for cultivating and processing Typha out of the text.

*This is a good point. We will mention the extra emissions for cultivation and processing of grass biomass harvest as well.*

L456: if the topsoil would have been stored under anoxic conditions, much more CH4 would have been emitted in CO2-equivalents than the 557 t CO2 per ha under oxic conditions, based on the papers of Harpenslager et al., (2015) and Quadra et al. (2023). The authors also mention this in line 468. So in that sense, the authors could be more positive, or less negative, about topsoil removal here.

*We were not necessarily negative about topsoil removal we only point out that there is a huge amount of carbon removed with the topsoil and this should be considered in the GHG balance. How much more $CH_4$ we would have gotten without topsoil removal is of course hard to say. We will mention the trade-off between potential higher CH4 emissions and CO2 reduction with respect to top-soil removal.*

L468: typo

*Typo will be corrected.*

L475-478: the highest chloride concentrations measured in the surface water were 62 mmol/l, which is equivalent to 2.2 g/l. This is in the range of the upper limit for T. latifolia and far under the upper limit of T. angustifolia. So the statements made here are not true.

*Indeed, the made statements are incorrect, thank you for noticing it, and will be rephrased as follows:*
*'... which is similar to the concentrations we observed and may partly explain the inhibited growth. For T. angustifolia our measured concentrations were lower than the upper limit of 7.2-8.8 g l-1 (Sinicrope et al., 1990).*

L482: typos.

*Typos not found.*

Conclusions: please rephrase based on the feedback given above.

---

## Author Comment (AC2)

Anonymous Referee #2, 03 Jan 2024

*We would like to thank Referee #2 for his/her effort to review our paper, and for the positive feedback, and points to improve or clarify. We agree with most of the comments that are given, and the suggested changes will help the paper to get to a higher quality and better readability. In the comments below we will elaborate on how we want to incorporate the suggested changes.*

General comments:

This is a comprehensive and well written paper which constitutes an important contribution about species-specific GHG balances from species relevant for paludiculture. The measurements have been conducted over a single year with limited frequency, which is common practice with this type of studies, but remains an important limitation. The study is ambitious as it aims to capture the fluxes of both CO2, CH4 (both diffusive and ebullitive fluxes) as well as important soil and water chemistry. There were also mentions of DOC, but these fluxes have not been reported. Inclusion of $N_2O$ would have strengthened the study further.

*We are happy to read the positive feedback about the relevance and quality of the paper. We agree that the measurement frequency is a limitation for the diffusive fluxes. The only other method that would overcome that problem would be the eddy covariance method but cannot be used on such a small scale as this experiment. One year of data is indeed also a limitation. We started a year earlier, but then the vegetation was not sufficiently developed, so we decided not to use the data. Extending the measurement period was financially not possible. DOC concentrations were indeed measured monthly, but since we do not know the exact water flow rate from the inflow ditch to the outflow ditch, we cannot calculate fluxes. $N_2O$ is indeed a missing GHG. With complete inundation of the soil, we do not expect much $N_2O$ from the paludiculture fields. From the reference site, $N_2O$ emission will most likely contribute significantly to the total GHG balance. This makes that we expect even a larger reduction of GHG emissions from the paludiculture fields. We will mention this in the discussion.*

The paper concludes that rewetting (flooding) with paludiculture reduces GHG emissions compared to an intensively used drained fen grassland, and that the choice of species is relevant for the success. Moreover, the paper discusses the pros and cons of topsoil removal at this site, and seem to suggest that topsoil removal may not have been positive here due to the removal of soil C, high CH4 fluxes, and possibly P-limitation for the plants. This is an important consideration for future studies as topsoil removal has been suggested to decrease CH4 emissions and limit P-leakage, without much consideration of where this soil should be stored to avoid continued soil C-oxidation off-site.

*We are happy to hear that the reviewer shares our concerns of topsoil removal.*

However, the study cannot determine the role of topsoil removal itself on post-rewetting GHG emissions as they have no reference site for this particular question.

*It is indeed a good point that we cannot exclude the single effect of topsoil removal on $CH_4$ emissions. The $CH_4$ emissions could have been even higher without topsoil removal, but we do not know. However, the fluxes we measured are in the upper limit of what we found in*

*literature from sites without topsoil removal, so it is the question if much higher emissions are realistic. We will address this point in the discussion.*

The study is also unable to answer questions about how GHG emissions from rewetting may be managed by the choice of water table depth, which asserts strong control on CH4 emissions in particular. This was briefly mentioned in the discussion but deserve further consideration due to its importance.

*We did study that too, together with different forms of manure applications, in the other basins (Figure 1C). We came to the conclusion that it would have been too much to elaborate on that in this paper, so these results are described in a separate paper (Vroom et al. in review). We will mention that we measured it and refer to the other paper more clearly.*

All in all, the study clearly answers the questions they set out to investigate, which is well-described by the title.

*We are happy to hear that.*

Scientific comments:

Material and methods:

There is a mention in the methods that Azolla died off and was replaced by *Lemna spp.* This is not discussed further anywhere. How fast did the Azolla deteriorate, was there x% left in September/October when the data in figure 6 suggest high concentrations of ammonium and total phosphorus in the pore water? And isn't that 'weakness' something worth mentioning when suggesting Azolla as a paludiculture species?

*We agree that this is not enough discussed yet.* Azolla *started to decline substantially from October (70% coverage by the end of October to completely disappeared in December), which does not completely coincide with the high ammonium and phosphorus pore water concentrations.*
*We will add the coverage of* Azolla *and* Lemna *spp. over time in the appendix and we will mention the risks of herbivory (for* Typha *as well) in the discussion.*

The reference site is presented in this section, and I wonder to what extent this site is representative of the conditions before topsoil removal and rewetting. The text does not discuss why this site was chosen, and the choice is not defended anywhere. Primarily the site is described to be different (intensively used, grazed, fertilized), hence my worry.

*We understand the concerns. The reference site is indeed not representing completely the situation before rewetting, since the land management is different at the reference site. However, the soil profile and water management are very similar. The reference site is a site that consists of conditions that are representative for drained peatlands in this area. We agree that it would be good to discuss this point and give a better description so that the reader better knows what the reference site represents.*

The study mention that diurnal fluxes were captured, but this data is not presented anywhere, which is unfortunate. There are several studies showing diurnal patterns of CH4 emissions from plants, but more data is needed to confirm which species this is relevant for. It would have been a nice addition to the supplementary material to visualize diurnal patterns, both if there are clear patterns or not. There is also no further mention if these diurnal patterns were included to interpolate data or to correct it.

*Indeed, diurnal patterns are expected, especially from the* Typha *species with the pressurized flow. We did look into this, but it is described in the other paper of Vroom et al.*

I am somewhat unfamiliar with bubble traps and would have liked a reference to the method to indicate if this is standard measurement practice.

*This method is used more often, and we will add a reference for the bubble trap.*

There seems to be no measurements of CH4 from the reference site, although emissions may have been negligible from the soil, they could have been rather large from ditches (if present). This is not discussed fully anywhere.

*There were $CH_4$ measurements done in the year before (2019) with a different chamber system (manual) and therefore we did not include it in this paper. The data are however described in a report. We refer to this report in the discussion.*
*We did not consider fluxes from ditches, also not at the paludiculture site, but only soil fluxes. We decided that that would be outside the scope of the study. We will mention this in the text.*

The authors use the term soil T when measurements are done in the 'soil' beneath a water column. It may be clearer to use the term sediment T when the site is flooded as in this case. I leave this to the authors' discretion, however.

*We agree that the term sediment T could also be used, but in our opinion soil T is more suitable as the material is not of sedimentary nature but rather inundated soil.*

The interpolation of CH4 based on soil T (or water T in case of Azolla) seems rather risky (which figure 7 clearly shows). The R-square is not high. Could it be possible to reach a higher R-square if more environmental parameters are included? This methodology is also not discussed, and no references to other studies applying this method are made. Is there a risk of an overestimation or underestimation of annual emissions?

*The method we used is based on the same idea as the temperature interpolation of ecosystem respiration ($R_{eco}$) with the Lloyd-Taylor function. With $R_{eco}$ we also know that there are more variables related than temperature (like vegetation). But we assume that the other factors are captured in the mean measured $CO_2$ fluxes (and thus also $CH_4$) every 2-3 weeks and are linearly changing over time. So we used the mean measured $CH_4$ flux per campaign, calculated this back to a reference temperature with the gained temperature relation (Fig. 7), linearly interpolated these reference emissions and then calculated the actual emission with the measured temperature and the temperature relation from this reference emission. This is not well explained in the text, so we will adapt the text to make this clearer. Furthermore, we will discuss this method by comparing it with other methods from literature in the Discussion section.*

Results:

It would have been interesting if the authors had supplied a simple RF-modeling to describe the GHG balances for the different species and the reference site (see Günter et al. 2020 – code freely available). This could also have included the C losses from the topsoil removal. I personally think that better visualizes and describes the net GHG impact from rewetting, where CH4 release is mitigated by CO2 uptake. However, I concede that in this particular instance, where measurements have only been made over a single year, with insect infestations etc. it may not be prudent to extrapolate emissions over several years (which is done in RF-modeling). This is perhaps something the authors could discuss though. Also, for future studies.

*Thank you for this suggestion. We were not aware of the model code, and we are very much interested in applying it to our dataset. If we decide that the results make sense, considering the restrictions of our dataset, then we will include it in the Results section 3.5.*

 Discussion:

The discussion is overall very good with much to consider and some helpful guidance to understand the results. I have mentioned some parts which are not discussed, which the authors may want to consider including for a strengthened paper.

*We are happy to hear that the discussion is considered to be very strong.*

I am very pleased to see that the authors mention that high DOC inputs may have influenced the CH4 emissions, but would also have liked to hear more about the authors' thoughts on its influence on CO2. Is it possible that NEE measurements were contaminated by allochthonous C? If the inflow had high concentrations of DOC, some of it may have been oxidized to CO2. Would the contamination be negligible or not?

*This is a good point that we have not really considered. We will mention in the discussion the possible overestimation of CO2 as well.*

The authors make a good point when they question to what extent (typo in text) the carbon storage in Typha will continue in the future. These studies should ideally cover more years than they frequently do…

*Typo will be corrected. The point of a study covering several years will be added to the discussion.*

I am very glad to see that the authors have included numbers on the potential C oxidation from the topsoil removal and how many years it would take to reach the same numbers from the reference site. This is often overlooked.

*We are happy to hear this point is received well.*

Technical issues

Abstract:

Please note that the species-specific CH4 fluxes do not match those in table 2.

*Indeed, thank you for noticing this. We will change this in the revised version.*

L30 "Azolla and T. angustifolia seem to have the highest potential in reducing…" (of these three species)

*Will be changed.*

L30 "complete rewetting" please consider using the term flooding.

*Will be considered.*

Introduction:

L43 The increase of CH4 emissions after rewetting depends primarily on the water table depth. This should be introduced here.

*We will introduce this.*

L43 "this gives an extra impulse" please reword to incentive.

*We agree with the rewording and change it accordingly.*

L44 "Rewetting 60%" This sentence does not describe what the reference presents. Clarify this statement. I.e. Rewetting 60% of the drained organic soils would turn the global land system into a net C sink by 2100, as opposed to a net C source as projected.

*We will rephrase the sentence as proposed.*

L51. It is quite possible that the degree of degradation (increased bulk density and thus higher SOC content per cm3) of the topsoil is important along with the nutrient status when it comes to the CH4 emissions. https://doi.org/10.1016/j.agee.2016.01.008

*Thank you for this additional reference, with more factors that relate to CH$_4$ emission after rewetting. We will include it in the introduction.*

Material and methods:

Figure 1B is very hazy. Is it possible to produce a map with higher dpi?

*We will increase the resolution.*

Equation 1, the minus sign within the brackets is difficult to see.

*We will add spaces next to the minus signs to make it more clear.*

Discussion:

L410 Please capitalize the two Wainscot bugs.

*We will do so.*

L452-453 Typha as insulation material, emissions…? I do not understand this sentence. Is it possible to clarify?

*We will change the sentence to: '…but if biomass is used sustainably for long term storage such as building material, this C-export should not be accounted for in the carbon/GHG balance.'*

---

## Author Response (AR1)

*General reply*

*We would like to thank Referee #1 and Referee #2 for their effort to review our paper and for the positive feedback, and points to improve or clarify. We agree with most of the comments that are given, and the suggested changes will help the paper to get to a higher quality and better readability. In the comments below we will elaborate on how we want to incorporate the suggested changes. Also, if we disagreed with the reviewer, we will elaborate on why we do not share their vision.*

*The most important changes we made are:*
- *The addition of radiative forcing calculations over time of the measured GHGs (as described in Günther et al. 2020). This give a better insight in the impact of topsoil removal and $CH_4$ emissions over time.*
- *We mention the measured $CH_4$ flux data of the reference site in the year 2019, and based on that, we assume that the $CH_4$ emission in 2020 would be zero.*
- *The gap filling method for the $CH_4$ diffusive fluxes is better described.*
- *Additional points were added to the discussion, like the missing $N_2O$ fluxes.*

*Reply Reviewer #1*

General: This manuscript describes an interesting case study on the effects of top soil removal, rewetting and the use of different wetland plant species on greenhouse gas emissions. The emissions fluxes were compared between these different plant species, but also with a reference grassland site. It would be nice to mention the total greenhouse gas balance in the abstract also, in CO2 equivalents, so that the reader can directly see the effect of the treatments on the greenhouse gas balance.

*Thank you for the suggestion. We have added the total GHG balances of the treatment and reference site in $CO_2$ equivalents to the abstract.*

Introduction:
L75. This is the first time that I hear that vegetated conditions may have higher CH4 emissions than non-vegetated conditions. Moreover, in the paper of Antonijevic it is stated that the period with elevated CH4 emissions ended with the occurence of cattail. So please correct that reference. And why are there no measurements of non-vegetated conditions in this experiment?

*There are many studies that show that vegetation leads to higher emission (Bodmer et al., 2024; Bastviken et al., 2023; Zhang et al., 2019; Hendriks et al., 2010; Kankaala et al., 2003) with the most important reason the carbon substrate input in the system for methanogens, and plant $CH_4$ transport. However, the oxygen transport to the rootzone also increases $CH_4$ oxidation, which in some cases leads to lower $CH_4$ emission (e.g. Vroom et al. 2018, van der Nat et al., 1998). But indeed, the study of Antonijevic was not the correct reference to back-up our point. We have changed the text by including the argumentation and references above.*
*We also conducted $CH_4$ measurements in non-vegetated conditions, but only with the manual chambers, so therefore we did not include it in this paper (but is described in Vroom et al. in review). These data also showed that the treatment without vegetation had the lowest diffusive $CH_4$ emission of all treatments.*

L95. No CH4 measurements at the reference site?

*CH$_4$ measurements were done on the reference site in a different year (2019) with a different chamber system (manual), therefore we did not include the results in this paper. However, we reconsidered this and mention the data in the methods and assume the CH$_4$ emission from 2020 would be similar (zero emission).*

Methods: are the two Typha compartments 430 m2 in total, or are they each 430 m2?

*They are 430 m$^2$ each. We have described this clearer in the methods.*

Figure 1C: indicate the inlet ditch and the water flow.

*In Figure 1 we aim to visualise the general overview of the field site and its geographic location. In Figure 2 the water flow, including inlet and outlet ditch is further described. We do not see added value in including these details in Figure 1.*

L135. Is it realistic to provide only inorganic fertilizer to the reference site? Does this give an underestimation of carbon fluxes to the atmosphere?

*The decision for inorganic fertilizer was made to prevent to have an extra carbon source for the carbon balance. If organic fertilizer was used, carbon input from organic fertilizer should be subtracted from the carbon and GHG balance. However, the carbon content of manure can vary to some extent, and it is unclear how long it exactly takes for the manure to be decomposed again. This results in higher uncertainties in the carbon and GHG balance. This uncertainty is not present with inorganic fertilizer, since there is no extra carbon input which needs to be corrected for.*

L180. It is a weak point that CH4 fluxes have appartently not been measured in the reference site. This flux could be zero of course, but then the authors should mention this. Also no N2O emissions were measured, which could have a major effect on GHG emissions, especially on the reference site. Please discuss the importance of N2O emissions somewhere in the introduction or discussion.

*As mentioned above as well, we will refer to the measured CH$_4$ emissions that are described in a report. As we measured a very small uptake of CH$_4$ , we assumed that the emissions in 2020 would be zero.*
*N$_2$O is indeed a missing GHG. With complete inundation of the soil, we do not expect much N$_2$O from the paludiculture fields. From the reference site, N$_2$O emission will most likely contribute significantly to the total GHG balance. Therefore, we expect even a larger reduction of GHG emissions from the paludiculture fields. We now mention this in the discussion adding the emission factor for N$_2$O for drained fertilized peatlands (L525-531).*

Results: it would be good to provide the actual biomass harvest values (per m2 or per ha). Now this is only mentioned in the discussion.

*We have added this to the results as well (L403-405).*

Fig.7 typo (And).

*Is removed.*

L370. Table 3. Figure 9. Why is all harvested biomass (C-export) considered as CO2 loss and thus as GHG flux? This totally depends on the biomass use. The grass from the reference site will partly be converted in CH4 by cows and the Typha biomass will for example only be converted to CO2 after a long time if it used as building or insulation material. This seems to be an important disclaimer here. The authors mention this in the discussion, but the disclaimer can also be mentioned here already.

*We have added the disclaimer about the use of the biomass in the results (L399-403).*

Discussion: how do Typha roots supply easily degradable carbon to the sediment? And is this in a significant order of magnitude to have effects on CH4 production?

*Roots lose carbon by root exudates, which is an easily degradable substrate. This is a very relevant carbon source for $CH_4$ production and thus emission (Bastviken et al. 2023). We have added this to the discussion (L461-462).*

L408-410: several typos.

*Typos are corrected.*

L410: I think that the damage to the T. latifolia plants is een important thing to mention, also in the abstract and conclusions, as it seems to be the reason for the very high methane emissions.

*Herbivory is already mentioned in the abstract and discussion as possible cause for the higher emissions. We have added it to conclusions (L569).*

L451-453: the authors mention the CO2 emissions for cultivating and processing Typha here, but do not mention the CO2 (and CH4) emissions for the reference site, i.e. the cultivating and processing of grass, milk, etc. This probably also (more than) compensates for the grass biomass harvest. So please make a fair comparison, or leave the statement about CO2 emissions for cultivating and processing Typha out of the text.

*We made the point so that it is clear that you do not gain much $CO_2$ reduction by using the biomass sustainably. It is not about comparing it to the reference site. So we left this point in without changing the text.*

L456: if the topsoil would have been stored under anoxic conditions, much more CH4 would have been emitted in CO2-equivalents than the 557 t CO2 per ha under oxic conditions, based on the papers of Harpenslager et al., (2015) and Quadra et al. (2023). The authors also mention this in line 468. So in that sense, the authors could be more positive, or less negative, about topsoil removal here.

*We were not necessarily negative about topsoil removal we only point out that there is a huge amount of carbon removed with the topsoil and this should be considered in the GHG balance. How much more $CH_4$ we would have gotten without topsoil removal is of course hard to say. We added now that removing only 5 cm instead of 20 cm would probably already have gained the desired $CH_4$ reduction (L523-525).*

L468: typo

*Typo is corrected.*

L475-478: the highest chloride concentrations measured in the surface water were 62 mmol/l, which is equivalent to 2.2 g/l. This is in the range of the upper limit for T. latifolia and far under the upper limit of T. angustifolia. So the statements made here are not true.

*Indeed, the made statements are incorrect, thank you for noticing it, and have been rephrased as follows:*
*'... which is similar to the concentrations we observed and may partly explain the inhibited growth. For T. angustifolia our measured concentrations were lower than the upper limit of 7.2-8.8 g l-1 (Sinicrope et al., 1990).*

L482: typos.

*Typos not found.*

Conclusions: please rephrase based on the feedback given above.

*Reply Reviewer #2*

General comments:
This is a comprehensive and well written paper which constitutes an important contribution about species-specific GHG balances from species relevant for paludiculture. The measurements have been conducted over a single year with limited frequency, which is common practice with this type of studies, but remains an important limitation. The study is ambitious as it aims to capture the fluxes of both CO2, CH4 (both diffusive and ebullitive fluxes) as well as important soil and water chemistry. There were also mentions of DOC, but these fluxes have not been reported. Inclusion of $N_2O$ would have strengthened the study further.

*We are happy to read the positive feedback about the relevance and quality of the paper. We agree that the measurement frequency is a limitation for the diffusive fluxes. The only other method that would overcome that problem would be the eddy covariance method but this cannot be used on such a small scale as this experiment. One year of data is indeed also a limitation. We started a year earlier, but then the vegetation was not sufficiently developed, so we decided not to use the data. Extending the measurement period was financially not possible.*
*DOC concentrations were indeed measured monthly, but since we do not know the exact water flow rate from the inflow ditch to the outflow ditch, we cannot calculate fluxes.*
*$N_2O$ is indeed a missing GHG. With complete inundation of the soil, we do not expect much $N_2O$ from the paludiculture fields. From the reference site, $N_2O$ emission will most likely contribute significantly to the total GHG balance. Thus, we expect even a larger reduction of GHG emissions from the paludiculture fields. We have mentioned this in the discussion and added the emission factor of $N_2O$ of drained fertilized peatlands (L526-531).*

The paper concludes that rewetting (flooding) with paludiculture reduces GHG emissions compared to an intensively used drained fen grassland, and that the choice of species is

relevant for the success. Moreover, the paper discusses the pros and cons of topsoil removal at this site, and seem to suggest that topsoil removal may not have been positive here due to the removal of soil C, high $CH_4$ fluxes, and possibly P-limitation for the plants. This is an important consideration for future studies as topsoil removal has been suggested to decrease $CH_4$ emissions and limit P-leakage, without much consideration of where this soil should be stored to avoid continued soil C-oxidation off-site.

*We are happy to hear that the reviewer shares our concerns of topsoil removal.*

However, the study cannot determine the role of topsoil removal itself on post-rewetting GHG emissions as they have no reference site for this particular question.

*It is indeed a good point that we cannot exclude the single effect of topsoil removal on $CH_4$ emissions, but that was also not the aim of the study. The $CH_4$ emissions could have been even higher without topsoil removal, but we do not know.*

The study is also unable to answer questions about how GHG emissions from rewetting may be managed by the choice of water table depth, which asserts strong control on CH4 emissions in particular. This was briefly mentioned in the discussion but deserve further consideration due to its importance.

*We did study that too, together with different forms of manure applications, in the other basins (Figure 1C). We came to the conclusion that it would have been too much to elaborate on that in this paper, so these results are described in a separate paper (Vroom et al. in review). We have mentioned that we measured it and refer to the other paper more clearly in the discussion (L555).*

All in all, the study clearly answers the questions they set out to investigate, which is well-described by the title.

*We are happy to hear that.*

Scientific comments:

Material and methods:
There is a mention in the methods that Azolla died off and was replaced by *Lemna spp.* This is not discussed further anywhere. How fast did the Azolla deteriorate, was there x% left in September/October when the data in figure 6 suggest high concentrations of ammonium and total phosphorus in the pore water? And isn't that 'weakness' something worth mentioning when suggesting Azolla as a paludiculture species?

*We agree that this is not discussed sufficiently yet.* Azolla *started to decline substantially from October (70% coverage by the end of October to completely disappeared in December), which does not completely coincide with the high ammonium and phosphorus pore water concentrations.*
*We have added the coverage of* Azolla *and* Lemna *spp. over time in the appendix (Fig. A1) and herbivory (for* Typha *as well) was already mentioned in the discussion.*

The reference site is presented in this section, and I wonder to what extent this site is representative of the conditions before topsoil removal and rewetting. The text does not

discuss why this site was chosen, and the choice is not defended anywhere. Primarily the site is described to be different (intensively used, grazed, fertilized), hence my worry.

*We understand the concerns. The reference site is indeed not representing completely the situation before rewetting, since the land management is different at the reference site. However, the soil profile and water management are very similar. The reference site is a site that consists of conditions that are representative for drained peatlands in this area. We agree that this should be better described and this has been done in the method section (L140-142).*

The study mention that diurnal fluxes were captured, but this data is not presented anywhere, which is unfortunate. There are several studies showing diurnal patterns of CH4 emissions from plants, but more data is needed to confirm which species this is relevant for. It would have been a nice addition to the supplementary material to visualize diurnal patterns, both if there are clear patterns or not. There is also no further mention if these diurnal patterns were included to interpolate data or to correct it.

*Indeed, diurnal patterns are expected, especially from the* Typha *species with the pressurized flow. We did look into this, but it is described in the other paper of Vroom et al which is mentioned in the text (L154).*

I am somewhat unfamiliar with bubble traps and would have liked a reference to the method to indicate if this is standard measurement practice.

*This method is used more often, and we have added a reference for the bubble trap (L183).*

There seems to be no measurements of CH4 from the reference site, although emissions may have been negligible from the soil, they could have been rather large from ditches (if present). This is not discussed fully anywhere.

*There were CH$_4$ measurements done in the year before (2019) with a different chamber system (manual) and therefore we did not include it in this paper. The data are however described in a report. We have reconsidered to use the data and now mention it (L104, L195-197, L416). There was a very small uptake of CH$_4$ found (close to zero) in 2019, therefore we assumed zero emissions in 2020.*
*We did not consider fluxes from ditches, also not at the paludiculture site, but only soil fluxes. We decided that that would be outside the scope of the study. We have mentioned it in the method section (L198).*

The authors use the term soil T when measurements are done in the 'soil' beneath a water column. It may be clearer to use the term sediment T when the site is flooded as in this case. I leave this to the authors' discretion, however.

*We agree that the term sediment T could also be used, but in our opinion soil T is more suitable as the material is not of sedimentary nature but rather inundated soil.*

The interpolation of CH4 based on soil T (or water T in case of Azolla) seems rather risky (which figure 7 clearly shows). The R-square is not high. Could it be possible to reach a higher R-square if more environmental parameters are included? This methodology is also

not discussed, and no references to other studies applying this method are made. Is there a risk of an overestimation or underestimation of annual emissions?

*The method we used is based on the same idea as the temperature interpolation of ecosystem respiration ($R_{eco}$) with the Lloyd-Taylor function. With $R_{eco}$ we also know that there are more variables related than temperature (like vegetation). But we assume that the other factors are captured in the mean measured $CO_2$ fluxes (and thus also $CH_4$) every 2-3 weeks and are linearly changing over time. So we used the mean measured $CH_4$ flux per campaign, calculated this back to a reference temperature with the gained temperature relation (Fig. 7), linearly interpolated these reference emissions and then calculated the actual emission with the measured temperature and the temperature relation from this reference emission. This is not well explained in the text, so we have adapted the text in the method section to make this clearer (L232-242). In this section more references are given for the temperature dependency.*

Results:
It would have been interesting if the authors had supplied a simple RF-modeling to describe the GHG balances for the different species and the reference site (see Günter et al. 2020 – code freely available). This could also have included the C losses from the topsoil removal. I personally think that better visualizes and describes the net GHG impact from rewetting, where CH4 release is mitigated by CO2 uptake. However, I concede that in this particular instance, where measurements have only been made over a single year, with insect infestations etc. it may not be prudent to extrapolate emissions over several years (which is done in RF-modeling). This is perhaps something the authors could discuss though. Also, for future studies.

*Thank you for this suggestion. We used the RF model and made long term estimates for the contribution of $CO_2$, $CO_2$ from topsoil and $CH_4$ emission to the radiative forcing. This is a very nice addition to our results (L428-443).*

Discussion:
The discussion is overall very good with much to consider and some helpful guidance to understand the results. I have mentioned some parts which are not discussed, which the authors may want to consider including for a strengthened paper.

*We are happy to hear that the discussion is considered to be very strong.*

I am very pleased to see that the authors mention that high DOC inputs may have influenced the CH4 emissions, but would also have liked to hear more about the authors' thoughts on its influence on CO2. Is it possible that NEE measurements were contaminated by allochthonous C? If the inflow had high concentrations of DOC, some of it may have been oxidized to CO2. Would the contamination be negligible or not?

*This is a good point that we have not really considered. We think, however, that this term is not that much compared to the overall flux. And since we suspect more an underestimation due to the high $CO_2$ uptake of carbon, we did not mention this point any further.*

The authors make a good point when they question to what extent (typo in text) the carbon storage in Typha will continue in the future. These studies should ideally cover more years than they frequently do…

*Typo is corrected. The point of a study covering several years is added to the conclusions (L568).*

I am very glad to see that the authors have included numbers on the potential C oxidation from the topsoil removal and how many years it would take to reach the same numbers from the reference site. This is often overlooked.

*We are happy to hear this point is received well.*

Technical issues

Abstract:
Please note that the species-specific CH4 fluxes do not match those in table 2.

*Indeed, thank you for noticing this. We have change this.*

L30 "Azolla and T. angustifolia seem to have the highest potential in reducing…" (of these three species)

*Is changed.*

L30 "complete rewetting" please consider using the term flooding.

*We changed the sentence to: …to have the highest potential in reducing greenhouse gas emissions after rewetting to flooded conditions*

Introduction:
L43 The increase of CH4 emissions after rewetting depends primarily on the water table depth. This should be introduced here.

*Is introduced.*

L43 "this gives an extra impulse" please reword to incentive.

*We agree with the rewording and changed it accordingly.*

L44 "Rewetting 60%" This sentence does not describe what the reference presents. Clarify this statement. I.e. Rewetting 60% of the drained organic soils would turn the global land system into a net C sink by 2100, as opposed to a net C source as projected.

*We have rephrased the sentence as proposed.*

L51. It is quite possible that the degree of degradation (increased bulk density and thus higher SOC content per cm3) of the topsoil is important along with the nutrient status when it comes to the CH4 emissions. https://doi.org/10.1016/j.agee.2016.01.008

*The suggested paper state the quantity of SOC can be a proxy for CH4 emissions. As suggested could the SOC content change with topsoil removal. However, our data show that with topsoil removal, the absolute amount of organic matter (OM) has not changed (see Table 1). Therefore, we have not include this reference in our paper.*

Material and methods:
Figure 1B is very hazy. Is it possible to produce a map with higher dpi?

*We have increased the resolution.*

Equation 1, the minus sign within the brackets is difficult to see.

*We have added spaces next to the minus signs to make it more clear.*

Discussion:
L410 Please capitalize the two Wainscot bugs.

*Is done.*

L452-453 Typha as insulation material, emissions…? I do not understand this sentence. Is it possible to clarify?

*We have changed the sentence to: '...but if biomass is used sustainably for long term storage such as building material, this C-export should not be accounted for in the carbon/GHG balance.'*

---

## Author Response (AR2)

*Dear editor and reviewers,*

*We would like to thank you for taking the effort for reviewing our manuscript again and come with some final points of improvement. We agree that the added radiative forcing to the Results deserved some better embedment in the manuscript. Therefore, we added text to the abstract and conclusions, and we discussed the results better. We also clarified the assumptions and changed the focus of radiative forcing a bit to avoid the direct comparison with the GWP. A detailed description of all the changes we made as requested from the reviewers can be found below.*
*Further, we have checked the entire document thoroughly for inconsistencies, typos and unclear descriptions, which we have changed/improved.*

*We hope you are satisfied with the changes we made and that the manuscript is ready for acceptance.*

*Kind regards (on behalf of all authors),*

*Thomas Gremmen & Merit van den Berg*

*Reply Referee#1*

Thanks for the revision of the manuscript. A lot of improvements have been made. I only have some considerations on the statements about top soil removal, a reaction on the new figure (10) and some minor comments:

*We are happy to hear that the referee is positive about the revisions.*

"L438-439: ""And we also assumed that the removed carbon from the topsoil will be decomposed to CO2 within 25 years.""
Please remove ""And"" at the start of this sentence.

*The sentence was combined with the sentence before, so now 'and' fits better.*

Furthermore, in L519-521 you state that ""If all the carbon that is removed from the top 20 cm (15.8 kg m−2) is not stored under anoxic conditions, an amount of 557 t CO2 ha 520 -1 will be released over the period needed to decompose that carbon. That is the same amount the reference site is emitting in 27 years.""
So the period needed to decompose that carbon is assumed to be 25 years (L438-439). Please also use this time period in L521 then. But why does it take 27 years to decompose that carbon at the reference site? What is the difference between the carbon in 20 cm of removed topsoil and the carbon in the topsoil of the reference site? I guess that the reference site (where the oxic layer is > 20 cm) emits more CO2 in 27 years than only 20 cm of top soil does in 25 or 27 years under oxic conditions? "

*In L521 we state that the amount of $CO_2$ that is expected to be emitted from topsoil removal is equal to the amount the reference site is emitting within 27 years. It is a bit uncertain how fast the topsoil will be decomposed, it depends on how much oxygen can enter the soil but we assumed more or less the same flux from the topsoil as from the reference site and rounded the number to 25 years. But to*

*prevent confusion about the numbers, we changed the decomposition period for the topsoil to 27 and explained that this is based on the flux from the reference site.*

*It could be that we underestimate the decomposition time of topsoil, since more carbon is exposed in the reference site. But the reference site is not losing young carbon but mainly peat, while in the removed topsoil net more young carbon will be decomposed, which has a higher decomposition rate than the recalcitrant peat.*

*In general, the precise decomposition time of the topsoil is not that important. The story would not change if the topsoil oxidizes within 20 or within 30 years.*

Figure 10: this figure is based on some assumptions (mentioned in the text), but what I miss is the assumption that biomass is harvested or not. If harvesting is assumed, what is assumed for the biomass application and possibly long-term carbon storage?

*We agree that we did not mention the point of harvest in these calculations. We assumed that all harvest is being decomposed to $CO_2$ again within the same year, the same as in Table 3 and Figure 9. We have written this in the text (L443).*

Next to that, nothing is mentioned about the radiative forcing in the abstract or conclusions now.

*We have added the results of the radiative forcing in the abstract (L34-36) and the conclusions (L587-589)*

L531: change "reference site" to "the reference site" or "reference sites"

*Is changed.*

L555: Change "grow better... ...below surface" to "...grow better at lower water tables, and CH4 emissions could be significantly reduced if the water table drops below soil surface."

*Is changed.*

L557: "...can be found in (Vroom et al., Under review)." Correct this.

*Is corrected.*

L567: Change "...with topsoil removal. Probably..." to "...with topsoil removal, probably..."

*Is changed.*

*Reply Referee#2*
Thank you for considering my review and the answers to my questions. You've done a great job revising the manuscript.

*We are happy to hear that the revisions are received positively.*

With the addition of the RF-modeling, I must however raise a concern. I find that this later addition is not thoroughly discussed in the manuscript. What you see in the results of the RF-modeling is that

all rewetting experiments lead to a period of warming compared to the drained state (curves do not cross the drained curve until 2080-ish). This period of time is important as it is relevant to the goals of the Paris Agreement.

It is also important to stress (also in the discussion) that the RF-modeling here builds upon a steady state (emissions do not change over time), which is not likely as you also mentioned in the discussion about possible reductions in CO2 uptake from Typha over time. It's also likely that these ponds will not be ponds for very long (the sediment and new peat will reach the water surface and perhaps above) which would possibly reduce CH4 emissions. I do not propose to change the input to the model, but to at least mention it in the discussion.

To address some of these concerns you could decide to discuss the RF-modeling merely as a way to highlight the differences between the species and the gases, and not necessarily the metric to use for climate mitigation purposes – as it is already internationally agreed upon to use GWP for this.

*We agree that some more discussion was needed for the RF modelling, and to point out the effect of a higher radiative forcing for the paludicrops on the short term and lower on the long term, compared to the reference site. We've changed the focus of the results to describe this effect and mention for each species the moment in time from when the radiative forcing of the paludicrops will be lower (L448-450). We removed the part where we describe the limitation of the static GWP (L436-438). We have also added a section in the discussion about the limitation of extrapolating this dataset to the year 2100 (1 year of data and expecting no changes in the fluxes) (L542-551). We do not think that these ponds will form new peat or that sedimentation causes the soil to be elevated above the water table. It is a highly managed (water) system where paludicrops are harvested and water table are kept artificially stable. So we did not add this point to the discussion.*

Details
Line 28. Typa (spelling) Typha

*Is corrected.*

Line 361 year (spelling) yearly budget

*Is corrected.*

Line 393 "resulting in the highest net ecosystem exchange (NEE) (Table 3)." Discussing NEE is difficult. Highest, although correct in that it was less negative, is a bit misleading. The exchange could be argued is higher if it is more negative than something negative, or more positive than something positive. It's the flux size, regardless of which side of the 0 line that's important here. I would suggest rewording this sentence.

*We agree that using the term 'highest' could be confusing with negative numbers. So we have changed it to: 'resulting in the lowest net uptake of $CO_2$ (NEE)'.*

Line 468 You changed to capitalize the Wainscot, but not the entire name. Should be Webb's Wainscot and Bulrush Wainscot

*Is changed.*